# Towards estimating the number of strains that make up a natural bacterial population

Tomeu Viver[1,2] ✉, Roth E. Conrad [3], Luis M. Rodriguez-R [4], Ana S. Ramírez [5], Stephanus N. Venter[6], Jairo Rocha-Cárdenas[7], Mercè Llabrés [7], Rudolf Amann [2], Konstantinos T. Konstantinidis [3] ✉ & Ramon Rossello-Mora [1] ✉

What a strain is and how many strains make up a natural bacterial population remain elusive concepts despite their apparent importance for assessing the role of intra-population diversity in disease emergence or response to environmental perturbations. To advance these concepts, we sequenced 138 randomly selected *Salinibacter ruber* isolates from two solar salterns and assessed these genomes against companion short-read metagenomes from the same samples. The distribution of genome-aggregate average nucleotide identity (ANI) values among these isolates revealed a bimodal distribution, with four-fold lower occurrence of values between 99.2% and 99.8% relative to ANI >99.8% or <99.2%, revealing a natural "gap" in the sequence space within species. Accordingly, we used this ANI gap to define genomovars and a higher ANI value of >99.99% and shared gene-content >99.0% to define strains. Using these thresholds and extrapolating from how many metagenomic reads each genomovar uniquely recruited, we estimated that –although our 138 isolates represented about 80% of the *Sal. ruber* population– the total population in one saltern pond is composed of 5,500 to 11,000 genomovars, the great majority of which appear to be rare in-situ. These data also revealed that the most frequently recovered isolate in lab media was often not the most abundant genomovar in-situ, suggesting that cultivation biases are significant, even in cases that cultivation procedures are thought to be robust. The methodology and ANI thresholds outlined here should represent a useful guide for future microdiversity surveys of additional microbial species.

A prokaryotic species is composed of multiple strains, the smallest distinguishable taxonomic unit within species, which typically show higher than 95% ANI among themselves vs. <90% ANI to strains of different species[1,2], revealing a natural gap or discontinuity in genome diversity at the species-level[3,4] ("clone" refers to identical genomes, and thus a strain may contain multiple clones[5,6]). Intermediate identity genotypes, for example, sharing 85–95% ANI, when present, are scarcer in abundance due to ecological differentiation, and thus should probably be considered distinct species[4,7] (for a contrasting opinion

that attributes such ANI gaps to cultivation or other sampling biases see[8]). While the sequence and gene-content diversity among strains of the same species is probably largely neutral[9,10], strains that carry unique gene content often underly the emergence of disease outbreaks[11,12] and/or the response of the species to environmental perturbations[7]. Therefore, a major challenge in microbiome research across environmental and clinical settings is to evaluate how many strains of the same species coexist in nature and how dominant strains emerge from this diversity in order to better quantify intra-species

diversity and understand how dominant strains emerge from this diversity. However, there is no appropriate, and/or widely accepted, definition of strain that can be used to answer these questions.

Bacteriologists have an operational definition of strain, which has its basis on the pure culture approach, and considers a strain as "*a group of genetically similar descendants of a single colony or cell*"[5]. Therefore, strain embraces all derivative lines of a single isolate, regardless of whether or not the descendants have undergone mutational events such as gene loss, duplications, genomic rearrangements, or modifications of the gene expression, as long as these do not affect the key (known) phenotypic properties of the strain (in cases that the mutational events involve a key phenotype of interest, the organisms may be split into multiple strains, but all are considered descendants of a single ancestor often called the wild-type). However, this concept is ambiguous because phenotypic similarity often depends on the growth conditions. For example, the isolation of an organism in the laboratory is commonly accompanied by changes in at least gene expression[13], and often gene mutations or deletions[14,15], due to adaptation to the laboratory conditions. Some of these changes could lead to substantial phenotypic differences; yet, the wild-type and the lab-adapted cells are typically considered the same strain[6]. In surveys of natural populations, where strain ancestry information is typically unavailable, strains have been discerned instead based on single nucleotide variants patterns (SNVs), but even in such cases a widely accepted definition on the number of SNVs expected to define a strain has not emerged yet[12]. Note also that strain should not be equated to 'clone' because the latter implies identical genetic sequence at selected loci or/and the whole-genome[6,16], which is not a prerequisite for members of the same strain.

To advance the current definition of strain, we used a large collection of isolates of the model hypersaline species *Salinibacter ruber* from two solar saltern sites in the Mallorca and Fuerteventura Islands, Spain[17]. Solar salterns are human-controlled tanks or ponds, used for the harvesting of salt for human consumption. These ponds are operated in repeated cycles of feeding with natural saltwater, increasing salt concentration due to water evaporation, and finally, salt precipitation. Several studies have shown that salterns in different parts of the world harbor recurrent microbial communities each year, characterized by low diversity of higher taxa (family-level and above), generally consisting of two major lineages i.e., the archaeal *Halobacteria* class and the bacterial class *Rhodothermia*[18], but with relatively high genus and species richness within each class[7,19,20]. Importantly, our previous studies have shown that *Sal. ruber* typically makes up about 5–10% of the total microbial community in most saltern sites across the world, including the two salterns sampled herein. Further, these abundant *Sal. ruber* populations typically harbor a large number of distinct genotypes[13,21], comparable to the diversity of genotypes of the model bacterial species, *Escherichia coli*, that can be found in the public databases and have been isolated from different sources[21]. Hence, salterns and *Sal. ruber* represent an ideal model system to study bacterial intra-species differentiation and units[17]. Here, the genome sequences of 138 *Sal. ruber* isolates, randomly chosen from our larger isolate collection, and the quantification of their diversity and in-situ abundance patterns using the metagenomes of the saltern of origin allowed us to propose a natural definition for a "strain" and other sub-species categories, as well as to evaluate the number of strains co-existing in their hypersaline ecosystem.

## Results

### Collection of isolates and identification of their clonal varieties (CVs) using PCR-amplicon fingerprint profiles (RAPD)

The isolates used in this study were recovered from two locations in Spain. The first collection was obtained from four adjacent ponds, fed with the same source seawater, in the 'Es Trenc' solar salterns on the Island of Mallorca during perturbation experiments performed over a period of one month in 2012. These experiments manipulated sunlight intensity through the application of shading mesh on the top of the ponds and salinity level through dilution with freshwater[7,22]. In total, 409 randomly picked pure cultures were isolated on standard growth media for *Salinibacter*[18,21], 207 of which were tentatively identified as *Sal. ruber* based on MALDI-TOF mass spectroscopy profiles[21]. A cost-effective and widely used approach to discern between putatively identical isolates (clones) is the random amplified polymorphic DNA (RAPD) PCR assay[23]. The isolates were dereplicated into clonal varieties (CVs) using up to three different RAPD primer sets. For this, we considered two isolates showing identical RAPD patterns as being members of the same clonal variety (CV) as also suggested previously[23] [note that the RADP primers do not target specific gene markers present in the genome but rather, they are short primers that can amplify/bind to several genome fragments and the resulting patterns of amplified DNA fragments are generally reproducible].

The 207 isolates represented 187 distinct CVs. Of them, 118 isolates, representing 107 CVs, including isolates from the same CV, were subjected to whole-genome sequencing. Therefore, these isolates represented both inter- and intra-CV diversity as well as the four ponds from the previous treatments (i.e., control, long- and short-shaded, and diluted; $n = \sim 30$ isolates/treatment)[21]. Note that since we did not necessarily aim to evaluate the existence of an intra-species ANI gap at least initially, the selection of isolates for sequencing intended to primarily increase diversity by largely sequencing members of different CVs and choosing CVs for this at random from the larger collection of CVs available, and secondarily to assess intra-CV diversity and highly identical genomes, which were therefore under-sampled. For instance, only four out of the total 9 CVs available with two or more representative genomes were included for sequencing and not all representatives of these CVs were sequenced. Accordingly, the 15 sequenced isolates represented four CVs with two or more representatives (Table 1 and Fig. S1), whereas the remaining 103 had unique profiles (single-isolate CVs). From the other location, the solar salterns on the Fuerteventura Island (Canary Islands), we obtained 46 isolates from a single sample of a salt-saturated (control) pond, 40 of which were identified as *Sal. ruber*. RAPD signatures identified 26 different CVs, and 25 isolates (56%) showed a unique CV whereas the remaining 15 showed identical RAPD profiles. Nine of the latter isolates were sequenced and found to belong to 2 CVs (CV5 and CV6; see below). In addition, 11 of the non-clonal isolates were sequenced and included in this study (Table 1 and Fig. S1).

### Comparisons of genomic relatedness among the isolates reveals an intra-species ANI gap

The collection of 138 *Sal. ruber* genomes recovered from the Mallorca and Fuerteventura solar salterns showed an average ANI value of 98.33% (SD = 0.37%) and a shared gene content of 73.38% (SD = 6.05%) (Fig. 1). ANI comparisons between isolates collected from the same sample showed similar results to the complete genome dataset (Fig. 1 and S2) and the Fuerteventura genomes did not cluster separately from the Mallorca genomes in the ANI space (e.g., Fig. 1), revealing that similar *Sal. ruber* populations were present in the two distantly located islands (central-east Atlantic vs. northwest Mediterranean Sea; ~2000 km geographic distance). From the 9,454 ANI pairwise comparisons among all genomes in total, 1.79% showed ANI ≥ 99.8%, 0.35% between 99.6% and 99.8%, and 4.04% comparisons fell between 99.0% and 99.6%. The majority of ANI values (93.81%) fell between 97.0% and 99.0%. Therefore, the distribution of ANI values revealed an intriguing pattern, with low frequency of ANI values between 99.6% and 99.8% accumulating only 0.35% of the total comparisons vs. 1.24% expected if the total ANI values higher than 99% (n = 578) were uniformly distributed between 99% and 100% (57.8 ANI values/genome pairs except per 0.1% bin vs. 15.5 values obtained); that is ~3.73 times fewer values than expected by chance alone (Fig. 1). By employing the kernel

**Table 1 | Statistics of the genome sequences used in this study and comparisons between genomes assigned to the same clonal variety (CV) vs. different CVs**

| Sample | CVs | Isolates | Nr. genomes (Nr. Subsampled genomes) | Genome size | | Num. CDs | | G + C content | | ANI | | % Genomic fraction shared | | Num. SNPs | |
|---|---|---|---|---|---|---|---|---|---|---|---|---|---|---|---|
| | | | | Average | stdev | Average | stdev | Average | stdev | Average | stdev | Average | stdev | Average | stdev |
| CZ | CV1 | CZ02, 06, 09, 10, 11 | 5 | 3,795,339 | 578 | 3144 | 2.55 | 66.06 | 0.001 | 99.9987 | 0.0004 | 99.53 | 0.22 | 106.7 | 52.74 |
| | | | (10) | 3,795,102 | 1379 | 3154 | 5.23 | 66.06 | 0.003 | 99.999 | 0.0005 | 99.50 | 0.03 | 96.6 | 57.93 |
| | CV2 | CZ01, 14, 15, 18, 19, 20* | 6 | 3,791,236 | 11,241 | 3169 | 57.03 | 66.06 | 0.013 | 99.9977 | 0.0021 | 99.24 | 1.17 | 230.07 | 226.58 |
| | | | (10) | 3,795,535 | 1,309 | 3158 | 4.12 | 66.06 | 0.002 | 99.999 | 0.0006 | 99.46 | 0.20 | 96.2 | 32.34 |
| | CZ unique | CZ03, 04, 05, 13, 20, 22, 27, 29, 30, 31, 32, 34, 35, 36, 37 | 15 | 3,644,174 | 92,085 | 3443 | 60.36 | 65.68 | 0.184 | 98.3352 | 0.2337 | 70.58 | 2.54 | 58,513.50 | 6,378.36 |
| CM | CM unique | CM02, 05, 06, 07, 08, 09, 10, 11, 12 | 9 | 3,823,282 | 69,250 | 3350 | 60.31 | 66.01 | 0.123 | 98.5508 | 0.3637 | 70.82 | 2.62 | 54,261.1 | 9340 |
| DZ | DZ unique | DZ01, 04, 05, 06, 07, 09, 10, 11, 12 | 9 | 3,630,009 | 91,461 | 3425 | 147.4 | 66.01 | 0.253 | 98.2728 | 0.2383 | 71.64 | 2.22 | 58,115.9 | 7089.9 |
| DW | CV3 | DW07, 11 | 2 | 3,710,853 | 623 | 3208 | 3 | 65.81 | 0.017 | 99.9991 | nd | 99.59 | nd | 107 | nd |
| | | | (4) | 3,710,023 | 348 | 3213.5 | 6.24 | 65.81 | 0.001 | 99.9991 | 0.0005 | 99.11 | 0.62 | 82 | 32.52 |
| | DW unique | DW07, 08, 10 | 3 | 3,688,086 | 17,366 | 3142.7 | 45.44 | 65.91 | 0.168 | 99.9666 | 0.0108 | 94.23 | 2.18 | 2,924.33 | 1188.76 |
| UZ | CV4 | UZ13, 14 | 2 | 3,879,012 | 392 | 3254 | 0.50 | 65.96 | 0.005 | 99.9994 | nd | 99.42 | nd | 82 | nd |
| | | | (4) | 3,878,572 | 924 | 3266 | 4.71 | 65.96 | 0.004 | 99.9991 | 0.0008 | 99.36 | 0.16 | 98 | 12.02 |
| | UZ unique | UZ01, 02, 03, 04, 05, 06, 08, 09, 10, 11, 12, 13 | 12 | 3,771,146 | 141,577 | 3642 | 169.52 | 65.47 | 0.230 | 98.2702 | 0.2615 | 69.61 | 4.33 | 60,896.40 | 8,666.32 |
| UM | UM unique | UM08, 10, 12 | 3 | 3,765,537 | 89,752 | 3335 | 120.33 | 66.01 | 0.292 | 98.26 | 0.5111 | 71,81 | 6.22 | 65,241 | 16,158.7 |
| FV | CV5 | FV6, 30, 23, 66, 69 | 5 | 3,732,326 | 13,119 | 3181 | 20.02 | 66.12 | 0.019 | 99.9976 | 0.0007 | 99.63 | 0.07 | 88.1 | 22.66 |
| | | | (10) | 3,728,750 | 10,824 | 3185 | 10.99 | 66.12 | 0.019 | 99.999 | 0.0003 | 99.61 | 0.06 | 81.6 | 28.12 |
| | CV6 | FV16, 35, 73, 68 | 4 | 3,811,847 | 1478 | 3250 | 2.21 | 65.97 | 0.002 | 99.9978 | 0.0006 | 99.57 | 0.07 | 103.67 | 51.12 |
| | | | (8) | 3,811,276 | 1621 | 3260 | 4,37 | 65,96 | 0.003 | 99.998 | 0.0031 | 99.61 | 0.09 | 91 | 18.85 |
| | FV unique | FV4, 5, 6, 8, 18, 41, 43, 53, 55, 60, 67, 70 | 12 | 3,802,109 | 59,614 | 3303 | 122.72 | 65.88 | 0.168 | 98.3352 | 0.0021 | 70.58 | 2.54 | 62,206 | 16,720.50 |

Rows that include "unique" in the second column represent comparisons between genomes assigned to different genomovars; numbers within parenthesis in the 3[rd] column represent the number of comparisons performed by randomly splitting the reads of a genome sequencing dataset into two halves (subsampling) and comparing the two halves (i.e., rows representing within genome comparisons); the remaining rows represent comparison between genomes assigned to the same genomovar. Samples CM, DZ, and UM yielded only isolates that each was assigned to a different CV and were not used for the subsampling calculations. Samples and isolates recovered from the Mallorca solar salterns are named according to the different ponds: control (C), dilution (D), and unshaded (U) at three time points: time-zero (Z), 1-week (W), and 1-month (M) (e.g., sample "UM" denotes Unshaded at 1 month)[21]. The sample and isolates from Fuerteventura area denoted by "FV" in the name.

*The genome was not used for subsampling due to low coverage
nd the standard deviation was not calculated given that only two genomes were representing the CV

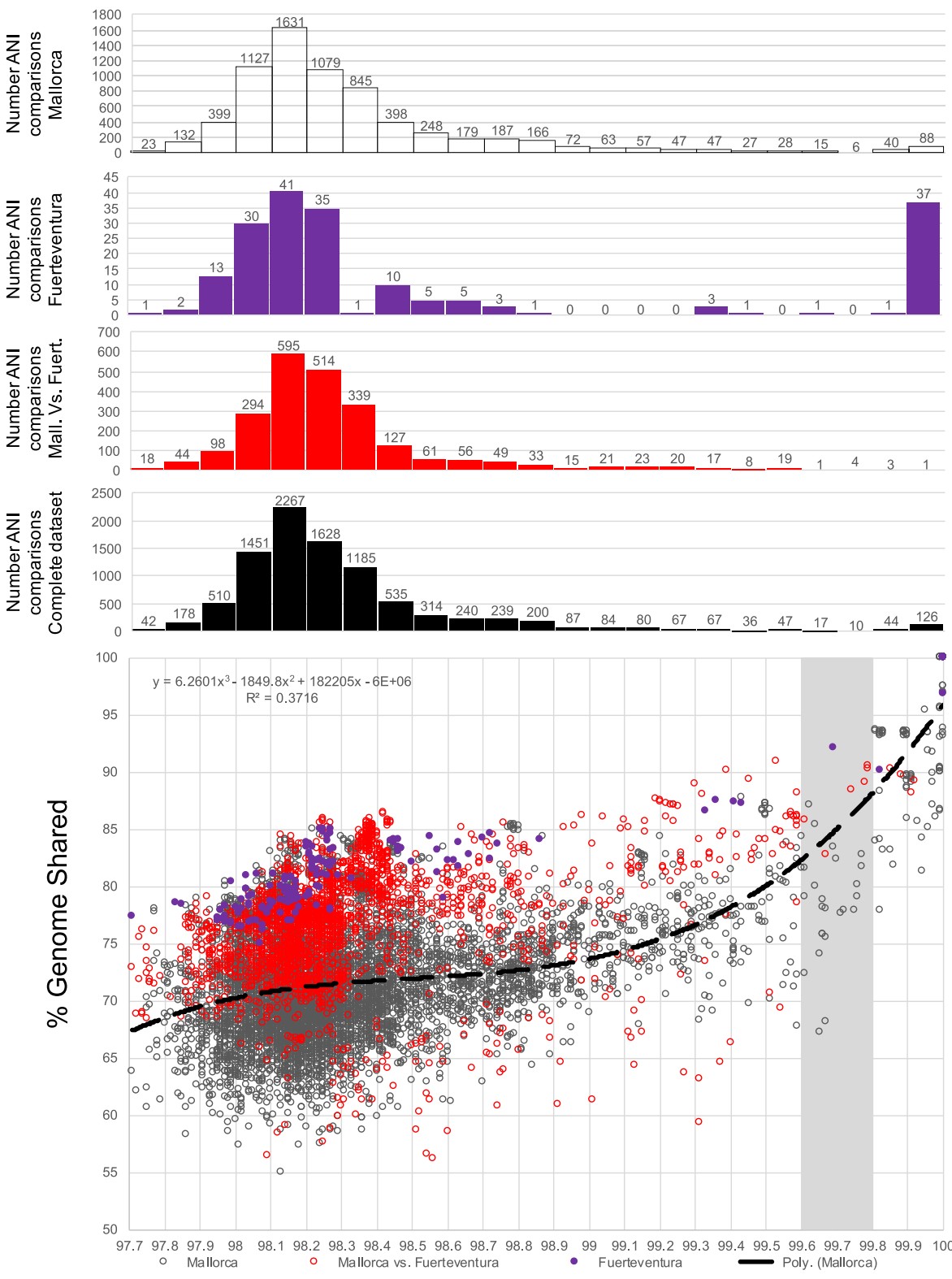

**Fig. 1 | Genomic diversity of *Salinibacter ruber* genomes used in the study in terms of ANI relatedness and shared genome fraction.** Each datapoint represents a comparison between two genomes and shows their ANI value (x-axis) against the shared genome fraction (y-axis). The graph on the top shows the number of datapoints for x-axis (in 0.1% windows or bins). Data points represent the 138×138 comparisons of our *Sal. ruber* isolate draft genome collection from Mallorca (118 genomes) and Fuerteventura (20 genomes) saline ponds combined (see also Fig. key for distinguishing datapoints by the place of isolation of the genomes compared). Note that the diversity of Fuerteventura genomes (in terms of ANI and shared gene content) is similar to that of many Mallorca genomes, albeit the latter collection also includes several more divergent genomes, in addition. Note also the shortage of data points (i.e., a gap) in ANI values around 99.6-99.8% (gray shaded area). Source data are provided as Source Data 1.

density estimate to statistically identify significant peaks and gaps in ANI data distribution, we observed a clear and sole ANI gap around 99.6-99.8% (Fig. S3), consistent with the visual inspection of the results mentioned above (gap here refers to the small number of genome pairs showing 99.6-99.8% ANI relative to counts of pairs showing ANI >99.8% or <99.6%). Bootstrap resampling analysis of the available genomes ($n$ = 10,000 replicates; Fig. S3) or calculating ANI based on core (shared) genes only (as opposed to all genes in the genome; Fig. S4) also supported the existence of the 99.6-99.8% ANI gap, although the gap was slightly shifted upwards in the core-gene ANI analysis. The latter result is expected because core genes tend to be more highly conserved, at the sequence level, compared to the genome average. Finally, including all available *Sal. ruber* genomes from the public databases, which increased our genome set to 211 genomes (from 138 genomes from Mallorca and Fuerteventura) did not affect the existence of the ANI gap (Fig. S5).

The shared gene content appears to follow ANI values, meaning that gene-content differences are typically larger among genomes with lower ANI values, but the relationship is bi-phasic. Genomes showing ANI >99.9% tend to have small gene-content differences (<5% of the total genes in the genome differ, on average). Differences in gene content increase sharply among genomes related between 99.5% and 99.9% before gene content differences stabilize around 15-35% for genomes showing <99.5% (Fig. 1). We examined the genomes that fell within the 99.6% and 99.8% ANI gap to gain a more quantitative view of the gene functions that differed at this level (Supplementary Note 1 and below for strain definition). Our evaluation showed that the latter genomes had substantial gene content differences that could underlie ecological differences and/or adaptations such as phage predation, albeit typically smaller than those observed between more divergent genomes (showing ~98% ANI) and larger than those observed between closely related genomes, ANI >99.8%.

## A natural definition for a genomovar and strain

The ANI gap revealed around 99.6–99.8% ANI (Fig. 1) was the only ANI range with such a strong bias in terms of pairwise comparisons not falling in this range. It is unlikely that this ANI gap is due to cultivation biases because the media and conditions used are thought to be robust for *Sal. ruber* and do not distinguish between members of the species or closely related *Salinibacter* species[18,21]. In fact, the ANI gap, most likely, is even more pronounced in nature because we did not sequence many isolates of the same CV as mentioned above (and almost all isolates of the same CV show >99.9% ANI among themselves; see also below). Therefore, the 99.6–99.8% ANI gap appears to be a property emerging from the data itself and to be robust. We suggest that the term "genomovar" could be used to refer to these ANI-based intra-species units. The term genomovar was originally used to name distinct genomic groups within species for which distinctive phenotypic properties have not yet been described and therefore, cannot be classified as distinct species based on the standard (low-resolution) taxonomic practices of the past[24]. The original definition was tailored "*to allow a nomenspecies to contain more than one genomic group, … analogous to other subdivisions of the nomenspecies*"[24]. The suffix -var indicates an intra-specific subdivision[5], not covered by the bacteriological code, but recommended to be used to describe intra-species groups. Hence, genomovar may capture well the intra-species ANI gap revealed by our analysis. We also propose the lower value of the ANI gap revealed ( ~ 99.5% ANI) as opposed to the upper value (99.8% ANI) of the gap as a more conservative threshold to define genomovars. For the strain level, 0.5% or 0.2% difference in ANI (correspondingly, 99.5% and 99.8% ANI) represents substantial, non-trivial, genomic divergence that, in most cases, would likely encompass several genomes with at least some phenotypic differences (due to substantial sequence and/or gene content differences among the genomes; see Fig. 1). Thus, multiple strains will be likely grouped together under the same 99.5% ANI

cluster in such cases, and strain, in general, represents a more fine-grained level of resolution than the 99.5% ANI level.

Accordingly, we propose to define strain as a group of isolates showing ANI values >99.99%. This threshold ensures high gene content similarly; e.g., shared genome usually >99.0% based on the *Sal. ruber* genomes analyzed that show >99.99% ANI, which is important for the current definition of strain that puts a lot of weight on (high) phenotypic similarity[10]. Further, this threshold encompasses well the typical sequencing and assembly noise observed. For instance, to quantify the noise resulting from the high-draft status (incomplete) of our genome sequences, we selected 23 isolates with sequencing depth >100X, split the raw data into two halves, and assembled the two subsets independently for direct pairwise comparisons of the resulting genome sequences. The ANI value between the paired re-assembled genomes showed values >99.99% and shared genome fraction >99.1% (Table 1; Sup. Data S1). Finally, a strain should not be equated to a clone as the latter implies an identical sequence, and the definition of strain may allow a certain degree of genome and gene-content divergence[24]. Consistent with this assumption, all but two genomes that were assigned to the same CV based on identical RAPD profiles also showed ANI values >99.99%. Thus, our proposed definition for strain is compatible with CVs as the latter have been traditionally defined, meaning a strain can encompass multiple CVs (discussed further below).

## Relative abundance of isolates in the samples of origin reveals cultivation biases

To assess the magnitude of isolation biases, we assessed what fraction of the total *Sal. ruber* population each isolate made up in the sample of origin, based on the number of metagenomic reads competitively recruited by the isolate-specific genes as well as (independently) the core (shared) genes. Given the high ANI identities among any isolate genome in our collection ( >97.8%) and the distribution of nucleotide identities of shared genes among the isolates (see also below), we defined the (total) *Sal. ruber* population in a sample as any metagenomic sequence from the sample sharing ≥95% nucleotide identity with any genome sequence in our collection [note that genomes sharing around 98% ANI may have several regions with lower sequence identity than their ANI value, but not much less than 95% for the orthologous parts;[1] hence, the 95% sequence identity threshold should capture the great majority of the *Sal. ruber* sequence diversity, albeit probably not all]. To assess the feasibility of our core gene-based approach, we first examined the nucleotide identity patterns among alleles of core-orthologous genes within vs. between genomovar comparisons. From the pangenome analyses, including all sequenced genomes from Mallorca and Fuerteventura (138 isolates), we detected 793 core single-copy orthologous groups (core-OGs: genes encoded in all genomes without paralogs). We considered an allele of a core gene to be unique when showing identity <99.8% to other alleles of the same gene based on the intra- vs. inter-genomovar ANI values, which also allowed for 1 to 2 sequencing and/or assembly errors resulting in SNVs that are common at the edges of assembled contigs. Our analysis revealed a total of 26,325 unique alleles among the 793 genes shared by our isolate genomes. About 66% of the core-OGs (523 out of 793) showed between 22 and 37 different alleles each (Fig. S6), and we identified two low-diversity core-OGs (high sequence conservation) that encoded for only ten alleles (ribosomal protein L32 and an ATP-binding cassette protein) and two highly diverse core-OGs with 111 different alleles each (cold shock protein and conserved hypothetical proteins). In general, all isolates belonging to the same genomovar shared always 100% of the core-OG alleles (Fig. S7) and an average of only 9.7% (SD = 14.5%) of the alleles were shared between different genomovars (Sup. Data S2). Further, and as expected, the clustering based on the percentage of shared alleles between genomes and the phylogeny based on the concatenated sequence alignment of the 793 core-OGs were in good agreement (Fig. S8). Therefore, it appears that

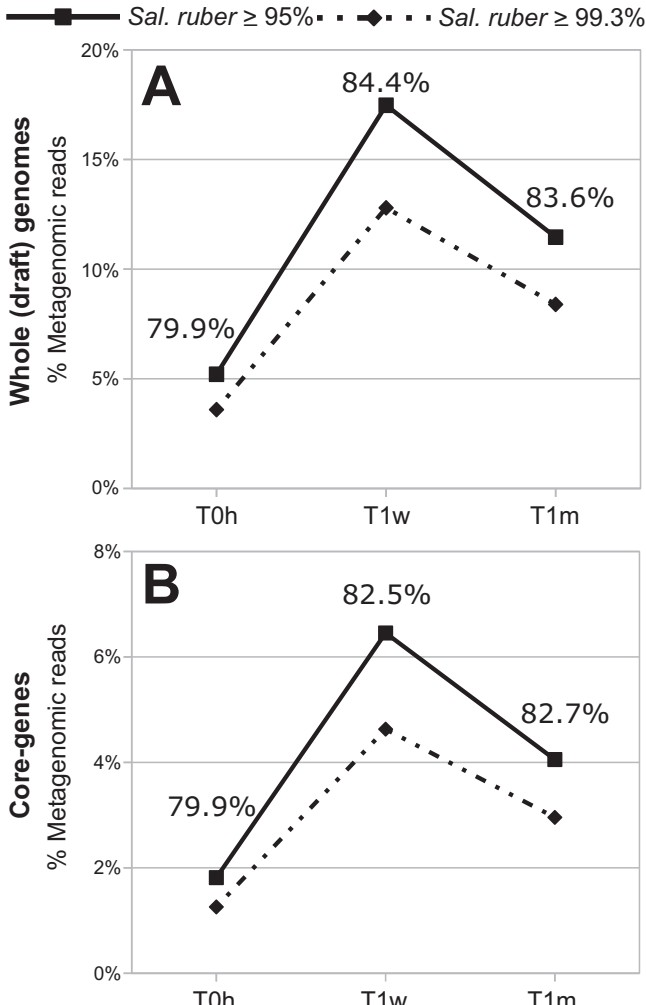

Legend above figure: ■— *Sal. ruber* ≥ 95%  ◆⋯ *Sal. ruber* ≥ 99.3%

**Fig. 2 | *Salinibacter ruber* abundance dynamics in the control pond during the sampling period of one month.** Metagenomic reads were aligned against *Sal. ruber* genomes using (**A**) whole (draft) genomes and (**B**) core genes only (see text for details). Continuous lines show the abundance of the natural *Sal. ruber* population (represented by reads mapping with identity ≥95% to any genome) and discontinuous lines show the percentage of reads represented by the sequenced isolates (reads mapping with identity ≥99.3% to any genome). Numbers for each time point indicate the proportion of total population reads that were associated to sequenced strains.

allelic variation in OGs between genomovars was generally adequate to distinguish genomovars (but not isolates or strains within genomovars), especially if metagenomic reads are recruited to representative genomes of genomovars (one representative per genomovar) competitively, albeit with some saturation in the signal (e.g., identical OG shared between genomovars). We dealt with ties in identity in read mapping by not counting such reads.

Interestingly, analysis of the results from competitive recruitment of *Sal. ruber* reads against representative genomes of genomovars showed that the most commonly recovered genomovars based on the number of isolates assigned to them were not necessarily the most abundant genomovars in the sample (Sup. Data S3). For instance, from the 16 different isolates recovered at time zero from the control pond of Mallorca, the most abundant genomovar-CV1 and genomovar-CV2 (represented by 5 and 6 isolates, respectively; genomovars are named after the most abundant, or only CV, its isolates were assigned to) were the least abundant in the metagenome (0.37% of the total the *Sal. ruber* population), and the isolates CZ13 and CZ27, each assigned to a

different single-isolate CV, represented the most abundant genomovars in-situ with 2.86% and 2.55% relative abundance, respectively. Similarly, from another pond (short-shaded pond at the initial time of the experiment, which is similar to the control pond because the shade treatment had not been applied to it yet[22]) two isolates were assigned to genomovar-CV4, which showed a relative abundance of 0.98%, while the most abundant genomovars in-situ were the (single-isolate) UZ12, UZ02, and UZ08, which showed a relative abundance of 3.03%, 2.76% and 2.62%, respectively. The results were similar between mapping to core vs. all genes in the genome (Fig. 2). The same analysis applied to the Fuerteventura metagenome revealed that the nine isolates represented by genomovar-CV5 and genomovar-CV6 (relative abundance of 10.6%) showed lower abundance than the (single-isolate) FV41 and FV43 (18.3% and 28.3%, respectively). Therefore, it appears that the number of isolates assigned to each CV is not a reliable proxy for the relative abundance of the corresponding CV in-situ, revealing a certain degree of isolation biases in our methodology, albeit most of the recovered CVs apparently represented abundant members of the in-situ population, making 0.1% or more of the total population (see also next section).

## Estimations of the total number of genomovars making up the natural *Sal. ruber* population

*Sal. ruber* showed a relative abundance ranging between 5.7% and 19.5% of the total microbial community represented in the control (no treatment) pond metagenomes of the Mallorca salterns, depending on the sample considered, and 6.12% of the total microbial community in the sequenced Fuerteventura metagenome sample. To estimate the level of coverage of the intra-population sequence and gene-content diversity achieved by our sequencing effort, we applied the read redundancy approach of Nonpareil[25] to the metagenomic reads identified as *Sal. ruber* (≥95% identity against any of the genomes in our collection) with a pairwise read identity threshold of ≥99.3% to identify redundant reads. The approach posits that if the reads are completely redundant, then the sequencing has saturated the extant intra-population diversity. We used 99.3% in this case to allow for one mismatch or sequencing error given our average read length of 150 bp, e.g., 149/150 = 99.33 [that is also the reason we cannot use short-read identity patterns to assess the existence of the ANI gap based on metagenomes because such reads do not offer resolution in the critical 99.6-99.8% nucleotide identity region]. Using 100% identity did not differentiate our conclusions substantially (e.g., we obtained a similar number of genomovars; see below). Note also that the 99.3% identity threshold is close to the 99.5% ANI threshold that distinguishes genomovars; thus, its use should not confound results from competitive read mapping against isolates of different genomovars. The estimated coverage achieved in the Mallorca control metagenomes when the three control samples were combined ranged between 93% and 95% (Table S1). These results indicated that our reads, collectively, sampled the great majority of the intra-population diversity of the *Sal. ruber* present in the samples, which was also consistent with the high relative abundance of the population in-situ reported above. Nonpareil's sequence diversity $N_d$ index also indicated that the intra-population diversity decreased only slightly during the sampling period and by the treatments, from 14.58 at time zero to 14.47 at time one month ($N_d$ is in natural log scale; Table S1).

Metagenomic reads mapping at high identity (≥99.3%) to an individual *Sal. ruber* isolate genome representing a distinct genomovar captured between 30% and 40% of the *Sal. ruber*-identified reads from Mallorca, depending on the sample and the isolate considered, and reflecting both genomovar-derived sequences as well as highly conserved genes across genomovars such as those originating from the rRNA gene operon (noise). Comparing the percentage of metagenomic reads assigned to the *Sal. ruber* species (identity ≥95%) and those mapping with high identity to genomovars represented by

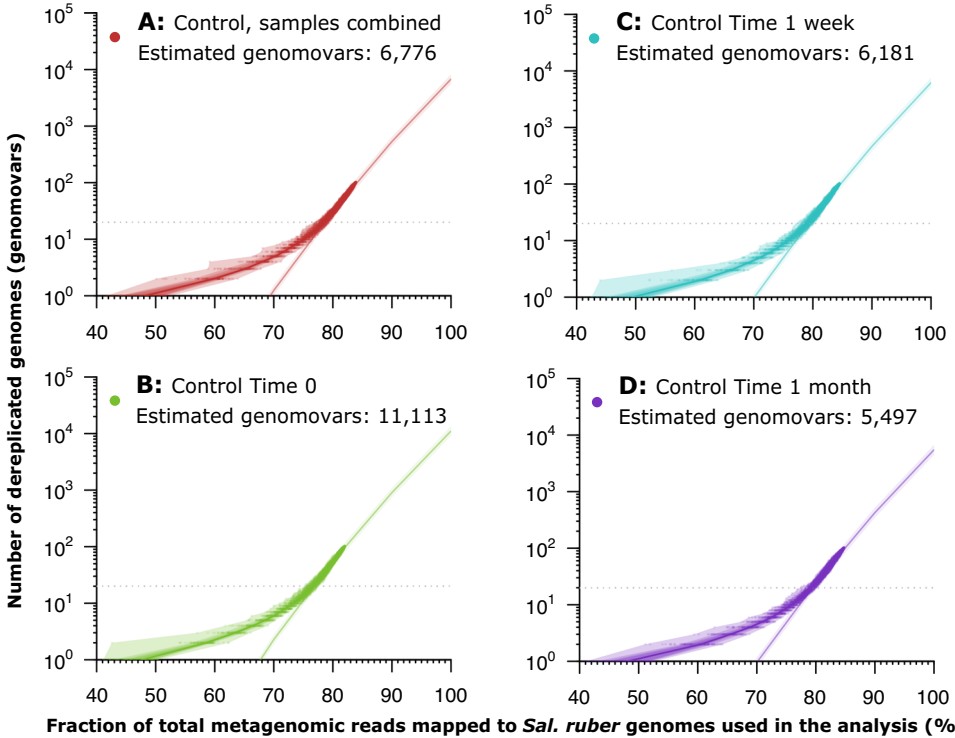

**Fig. 3 | Estimation of the number of genomovars making up the natural Salinibacter ruber population.** Metagenomic reads from each sample (Panels **B**–**D**) or all samples combined (Panel **A**) of the control pond were mapped to the *Sal. ruber* genomes preserving all matches with identity ≥99.3%. The mapping file was manipulated to remove one target genome at a time (randomly sorted) while recording the number of unique reads mapping at each step, and this process was repeated 100 times to reduce the impact of randomization on the estimates obtained (below). The number of reads were then expressed as the fraction of the maximum number of reads from the *Sal. ruber* species by dividing the observed counts by the total number of reads mapping to any reference genome with identity ≥ 95%. The logarithm of the number of total (dereplicated) genomes used was then expressed as a function of the fraction of *Sal. ruber* reads captured by the genomes, and a linear regression was determined by unweighted least squares and evaluated using Pearson correlation for the region between 20 and 100 genomes. This trendline was extrapolated to 100% coverage of the genomovar diversity (i.e., all reads from the species) to provide an estimate of the number of genomovars represented (Y-axis) in the total sequenced fraction (X-axis). Filled dots represent the fraction of the total *Sal. ruber* reads captured by the genomovars used, and the shaded bands around the observed subsamples represent the central inter-quantile ranges at 100%, 80%, 60, 40%, and 20%. Source data are provided as Source Data 2.

isolates in our collection (identity ≥99.3%), our collection of sequenced isolates recruited 79.9% of the species reads at time zero, 84.4% at one week, and 83.6% at one month of the corresponding samples from the Mallorca control pond (salt saturation conditions). Similarly, the 12 sequenced genomovars (20 isolate genomes, in total) represented a total of 77.3% of the metagenomic reads assigned to *Sal. ruber* for the Fuerteventura metagenome (Fig. 3). Therefore, about 20% of the reads assigned to *Sal. ruber* were not recruited by our isolate collection, indicating that a higher effort in cultivation is necessary to fully recover all abundant genomovars of the population. Nonetheless, these results also showed that our isolate collection collectively represents the majority (and therefore, the abundant members) of the in-situ *Sal. ruber* population.

To evaluate the number of genomes that will be necessary to sequence in order to recover the full in-situ species genome diversity (at the genomovar level, since we cannot reliably resolve between genomes or strains of the same genomovar) using the same isolation procedure as applied herein, we built a rarefaction curve of the number of reads recruited by each isolate against the number of isolates included in the analysis (one isolate genome per genomovar was used in the analysis to avoid the effect of ties in read mapping against very similar genomes). We observed that, using a logarithmic scale (Fig. 3) between 20 and 100 genomes, the rarefaction curve is almost a straight line (Pearson correlation value of 0.99935), and we would need to isolate and sequence around 11,000 isolates (99% prediction interval: 9205 – 13,416), each representing a distinct genomovar, in the more diverse sample (control time zero) and around 5500 isolates

(4501 – 6711) in the lower diversity sample (control time 1 month) in order to capture the complete extant genetic diversity (Fig. 3 and Table S2).

## RAPD genotyping is a reliable method for identifying redundant genomes or members of the same strain

The available genomes allowed us to assess the resolution of the RAPD method for identifying clonal isolates based on direct comparisons to the genome sequences of the corresponding isolates. In general, we observed that almost all isolates sharing identical RAPD patterns using three different PCR primers also showed ANI values >99.99% and shared genome >99.24% while genomes of different RAPD patterns (different CVs) typically showed ANI values <99.8% and shared genome <96%. Accordingly, the genome size between isolates of the same CV varied only slightly, between 392 bps in CV4 and 13,119 bps for CV5 (Table 1; Sup. Data S4). We identified only two exceptions to these patterns. Specifically, although the nine sequenced isolates from Fuerteventura solar salterns displayed an identical RAPD pattern, we decided to split the genomes in two different CVs (CV5 and CV6) based on genomic differences (i.e., ANI were <99.90% and 70 genes consistently differed between isolates of CV5 vs. CV6). Further, although the isolates assigned to CV1 and CV2 showed different RAPD patterns based on RAPD4 primers, their genomes displayed very high similarity, comparable to that observed in genomes within other CVs (Sup. Data S5). The ANI between members of CV1 and CV2 was 99.99% on average (SD = 0.0012%) and percentage of shared genomic fraction 99.38%, on average (SD = 0.41%). We were not able to detect any substantial

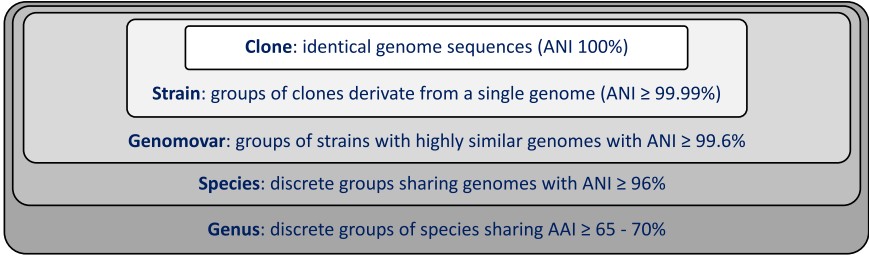

**Fig. 4 | Proposed thresholds to define species and intra-species units.** Note that thresholds are based on average amino-acid identity (AAI) for genus level[33] and average nucleotide identity (ANI) for species[1,2,4] and intra-species level.

difference based on the genome comparison and thus, merged the two CVs into one CV. In summary and based on our proposed definition for strain (>99.99% ANI and >99.0 shared gene-content) and analysis of 138 isolates genomes, in >95% of the cases the RAPD methodology was accurate in assigning a genome to the same (or different) strain based on identical (or non-identical) profiles. Thus, RAPD profiling represents a quick, cost-effective and reliable method for genotyping isolates based on genomic evaluation, which is consistent with previous results based on low-resolution fingerprinting methods[24].

## Discussion

Distinct units within bacterial species have been recognized long time ago, often driven by technological advancements providing higher resolution, and have been designated with different names such as subspecies, ecotypes, clonal complexes, serotypes, pathovars, phagovars, genomovars and strains, among several other designations[26] depending on the geno- or phenotypic traits used to discriminate among them. However, the standards applied to define each of these units have commonly been inconsistent between different studies, and the units are often not applied based on the same means (e.g., marker genes used) across different bacterial taxa, creating challenges in communication about diversity. In a comparison of 138 *Sal. ruber* isolates we observed an area of genomic discontinuity between 99.6% and 99.8% ANI that accumulated only about one quarter of the total pairwise measures expected by chance in case of a uniform ANI value distribution. This natural gap emerging from the data itself, as opposed to a manmade arbitrary threshold, is statistically robust (e.g., Figs. S3 and S5). Further, it is important to note that this ANI gap may be even more pronounced in-situ than our isolate collection indicates because we opted to cover diversity (e.g., sequencing a single member of many different CVs) rather than to sequence highly related genomes of the same CV. Indeed, isolate sequencing included only four of the total nine CVs with two or more members, and not all members available for these CVs but rather a randomly chosen subset. This gap led us to suggest the concept of genomovar[24] for all organisms sharing genome-wide ANI >99.5%. Our proposal provides a more accurate and standardized definition of this unit compared to previous practice. Importantly, a recent analysis of 330 bacterial species with at least 10 complete (not draft) sequenced isolates per species available in the NCBI database, as well as long-read metagenomes from various habitats, showed that the 99.2-99.8% ANI gap is also found within most of these well-sampled species (at least 70% of them)[27]. Therefore, the genomic discontinuity around 99.5% ANI observed here based on the *Sal. ruber* genomes may be a more broadly applicable property of bacterial species, and thus a reference point for (more naturally) defining genomovars within species. We also suggest evaluating the ANI value distribution for the species of interest, and if the data indicate so, to adjust the ANI thresholds proposed here to match the gap in the observed ANI value distribution.

Our analysis also revealed that there is significant ANI and gene content diversity within a genomovar, defined at the 99.5% ANI level. To circumscribe this diversity, we also propose ANI >99.99% as a

general purpose and practical threshold to define a strain within a genomovar, which ensures high gene content similarity among members of the same strain (typically, >99.0% of total genes in the genome), an important prerequisite of the current definition for strain[5]. This threshold is also robust to the typical noise and artifacts emerging from the genome sequencing and assembly steps based on our analysis (Table 1). Further, this threshold almost always corresponded to identical RAPD fingerprints, with only a few exceptions of genomes sharing >99.99% ANI that differed in their RAPD profile and/or the gene content substantially. For example, isolates of the CV6 RAPD group could be differentiated from CV5 isolates by just the presence of a plasmid in the former, which encoded for a CRISPR-Cas system thus, could provide an adaptively immune function against viruses[28,29]. We propose to consider such cases of genomes having substantial gene-content differences attributed to mobile elements while sharing >99.99% ANI still as representative of the same strain; in other words, to let the ANI >99.99% threshold override mobile element differences in order to simplify strain identification and communication. Further, mobile elements are often ephemeral and do not confer substantial phenotypic differences in cases that they are not expressed or carry functionally important genes. However, in cases where important phenotypic differences that distinguish between organisms sharing ANI >99.99% are known such as antibiotic resistance genes carried by plasmids, our proposed definition for strain could be neglected or adjusted to even higher ANI values as seen appropriate. It follows that a strain encompasses the concept of a clone that implies identical genome sequence (Fig. 4).

Based on the abovementioned definitions and thresholds for genomovar and strain, our findings suggest that the number of (distinct) genomovars making up the *Sal. ruber* population appears to be large but not unfathomable. Specifically, our mathematical extrapolations indicated that to completely cover the genomic diversity of the natural population of *Sal. ruber*, we would need to isolate and sequence between 5,500 and 11,100 isolates, each representing a distinct genomovar; a large but bounded number. These results contrast with a previous estimation indicating that diversity within species may be unbounded[30]. It should be mentioned, however, that since our sequencing effort did not saturate the diversity within the natural *Sal. ruber* population (~95% of diversity was covered by sequencing), it is possible that our estimates on the number of isolates needed might change with higher coverage, especially if the diversity not captured by our sequencing effort harbors a disproportionately lower (or higher) number of genomovars than are accounted by our fitted trendlines (e.g., Fig. 3). Furthermore, performing this type of analysis with long-reads, or even single-cell amplified genomes, could reduce uncertainty in the estimations caused by the use of short reads, which cannot resolve well very closely related genomes due to the limited sequence available. Most importantly, such long-read data will reveal whether or not the associated sequence thresholds for identifying redundant reads by our study underestimated the diversity of genomovars within the *Sal. ruber* population. Despite these technical limitations, however, the results and approach outlined here should represent a useful

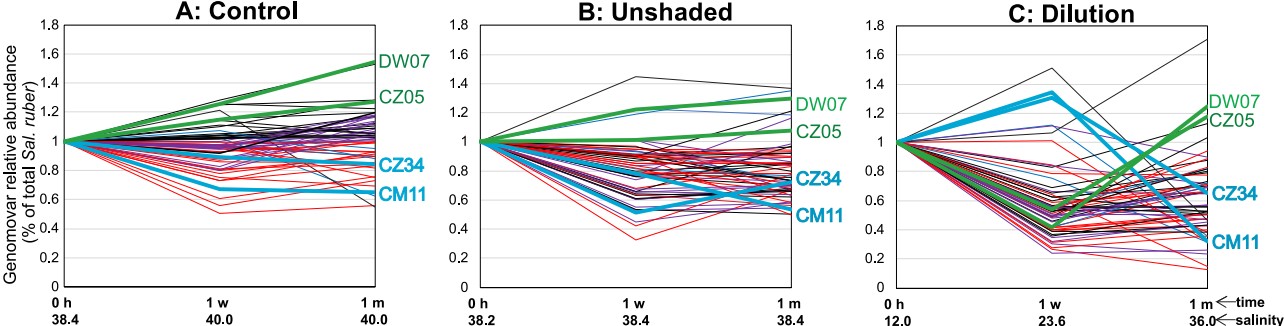

**Fig. 5 | *Sal. ruber* genomovar abundance dynamics over the one-month period of experimental manipulation of sunlight intensity and salinity.** Each line represents a (distinct) genomovar and shows its relative abundance as a fraction of the total *Sal. ruber* population, based on the number of metagenomic reads uniquely recruited by the representative genome of the genomovar (y-axes), against the three metagenomic sampling time points (x-axes) for each of the three separate experimental ponds used (panel title on top). Lines are colored in black or red if the corresponding genomovar increased or decreased in abundance in the control pond (Panel **A**), respectively, except for four genomovars (denoted on the panels) that showed significant difference in abundance in the dilution (Panel **C**) relative to the control pond (same color is used for the same genomovar across panels). Note that, for the dilution pond, the salt concentration was reduced from 33.6 to 12.0% by the addition of freshwater at time zero (0 h); the unshaded pond (Panel **B**) was kept uncovered -like a control pond- until 0 h and, after the first sample was collected, was covered with a shade mesh that reduced sunlight intensity by 37-fold for one month, as described in detail previously[21]. Source data are provided as Source Data 3.

reference point for future studies that aim to quantify intra-population and intra-species diversity in *Sal. ruber* or other species.

In contrast to the high genomovar diversity estimated, we find it remarkable that our relatively small collection of sequenced isolates (107 out of 187 unique RAPD profiles for the Mallorca ponds) already represented ~80% of the *Sal. ruber* population in the habitat, on average (Fig. 2). These results suggested that most genomovars not represented by our isolates had very low abundance in-situ, at least at the time of our sampling. Therefore, it appears that the concept of the rare biosphere for the species or population levels[31] applies equally well to the genomovar abundance patterns within the natural *Sal. ruber* population. These results also suggested that our isolation procedures were overall robust and did not dramatically bias the diversity of the *Sal. ruber* population recovered, meaning that the genomovars that appear to be the most abundant in-situ were represented among the recovered isolates. It should also be mentioned that the saltern ponds are never fully emptied and thus, the high genomovar diversity observed may be due, at least in part, to the existence of a persistent inoculum of (diverse) strains at the start of each water-overflowing/evaporation/salt-precipitation cycle (e.g., strain diversity is not re-assembled de-novo from surrounding environments). Collectively, these results also suggested that *Sal. ruber* is not an "outlier" with respect to its intra-species diversity patterns but simply a well-adapted species to its respective environment.

Despite the overall high representation of the *Sal. ruber* extant diversity by our isolates, however, the most frequently recovered genomovars from a single sample were frequently not the most abundant genomovars in-situ. In fact, it appeared that the most abundant genomovars by isolation often were not the most abundant in-situ genomovars among those represented by our isolate collection based on competitive read mapping to isolate/genomovar-specific or core genes. This result might not be surprising given that cultivation biases are well appreciated in microbial ecology for several decades now[32] and most *Sal. ruber* genomovars or strains recovered appear to be relatively low-abundance in-situ (but not rare since each genomovar typically recruited a substantial number of unique reads) according to our analysis (Fig. 3). Nonetheless, our results do show that the laboratory growth conditions could select for less abundant members of a population (i.e., lead to cultivation biases), even in cases where the growth media may be specifically-designed for the target population and appear to be robust, in general. Therefore, for reliable estimates of in-situ abundance, isolation efforts need to be combined with culture-

independent data even in cases where growth media are thought to not be restrictive for the target organism. Recovery of a higher number of isolates could also help with cultivation biases, e.g., despite our substantial efforts only a small number of isolates were members of the same genomovar due to the high intra-species diversity, albeit obtaining a higher number of isolates than what was achieved here ($n = 409$) may be impractical and/or costly. Nonetheless, these cultivation biases are unlikely to have affected our key results (e.g., existence of the ANI gap) and conclusions for the reasons mentioned above such as that our isolates largely represented the abundant members of the natural *Sal. ruber* population and were randomly chosen for sequencing from a larger collection. Notably, we have recently observed a highly similar ANI genomovar gap for many other species and based on long-read metagenomes[27], as mentioned above. Despite our efforts to account for cultivation biases as much as this was possible, and our data indicating that intermediate genotypes whose identity falls within the ANI genomovar (and species) gap -when present- are likely due to ecological differentiation (see[4] and below), we can not exclude, at present, the possibility that cultivation and/or sequencing biases might have accounted for some of the results reported here. Sequencing of more genomes of the same species from these salterns (and other habitats) in the future would be important to further corroborate the findings reported here.

A remaining question is what ecological (e.g., selection, drift) and/or genetic (e.g., recombination frequency) mechanisms may underly the existence of distinct units (e.g., genomovars) and strains within species. Our preliminary results indicated that at least some of the genomovars may not be redundant among themselves (i.e., they do not represent neutral diversity) but instead, show distinct ecological preferences. Specifically, we noted that while genomovar relative abundance during the one-month duration of the experimental manipulation did not change substantially, and/or changed stochastically, in the control (no treatment) and unshaded ponds (the pond was kept uncovered, like a control pond, until the time zero sample was collected and, immediately after, was covered by a shade mesh for one month; i.e., short-shaded), the abundance of several genomovars varied much more dramatically during the manipulation of the in-situ salinity concentration by the addition of freshwater (dilution pond; Fig. 5; note the larger fluctuations in the abundances of several genomovars in panel C vs. panels A and B). For example, genomovars CZ34 and CM11 appear to prefer relative low salinity concentrations (12 to 23% salts) and decreased in abundance when conditions reached salt-

saturation level (~36% salts) due to sunlight-driven water evaporation, while genomovars DW07 and CZ05 showed the opposite abundance pattern. Genes related to osmoregulation are also enriched in the genome of the latter genomovars, consistent with our previous gene-centric analysis of the corresponding metagenomes[21], indicating that high-salt preference and/or tolerance may distinguish functionally, and thus ecologically, at least some of the genomovars. Nonetheless, representative isolates of all these genomovars were recovered from the control as well as the treatment ponds. The latter results indicate that while the ecological advantage of genomovars DW07 and CZ05 at high-salt conditions may be significant, it is apparently not strong enough to purge diversity within the species (i.e., completely out-compete remaining genomovars), and genomovars that may be less-adapted to high-salt concentrations (e.g., CZ34 and CM11) appear to survive such (unfavorable) conditions, presumably as (viable) members of the rare biosphere. While this hypothesis remains to be tested experimentally, it does provide a plausible explanation for how this immense intraspecies genomovar and strain diversity is maintained. That is, members of the same genomovar share a more similar ecological niche than members of different genomovars, and this ecological cohesiveness also keeps the former genetically more similar to each other. We have also found that genomovars with low-salt (or high-salt) preference are distributed across the *Sal. ruber* phylogeny, i.e., they are not clustered together in a major clade (Fig. S8). These results also indicated that the corresponding ecologically important genes related to salt sensing and tolerance are likely moving horizontally between genomovars frequently, a hypothesis that should be tested more rigorously in the future.

The thresholds proposed here (Fig. 4) to define genomovars and strains provide convenient and practical means to define these sub-species units and thus, should greatly facilitate future studies in environmental or clinical settings. Although we consider our results with *Sal. ruber* to be relevant for many *Bacteria* and *Archaea*, it is likely that other species with different lifestyles and/or dispersal strategies show different diversity patterns. Our methodologies and results on the diversity and number of genomovars found within a natural population as well as the extent of cultivation biases should represent a useful guide for future microdiversity surveys of other species.

## Methods

### Experimental description

The *Sal. ruber* isolates analyzed here were collected from six adjacent solar saltern ponds located in Mallorca (Spain) that have been previously used for manipulative experiments described in ref. 7,22. In addition, as part of the present study, we included one metagenome sample from Fuerteventura saltern (Salinas del Carmen) located in Canary Islands (Spain) collected in 2020 and 40 isolates that originated from this single sample. The Fuerteventura sample was collected in accordance with the permit ESNC27, with the unique identifier ABSCH-IRCC-ES-241224-1 that has been provided by the Dirección General de Biodiversidad y Calidad Ambiental del Ministerio para la Transición Ecológica of the Spanish Government.

Isolates from these samples were recovered in sea water medium supplemented with 25% salt concentration[34] and taxonomically identified by Matrix-Assisted Laser Desorption Ionization-Time of Flight Mass Spectrometry (MALDI-TOF MS) as described in ref. 35. RAPD fingerprinting[36] was applied to identify clonal isolates. RAPD fingerprints were obtained using the RAPD4, RAPD5, and RAPD6 amplification primers[23].

### Genomic sequencing and analysis

DNA extraction was performed as detailed ref. 37. DNA sequencing libraries were prepared using the Illumina Nextera XT library prep kit and libraries were sequenced using a NovaSeq6000 150PE (2 x 150 bp) instrument. Raw reads were trimmed, and quality filtered to remove low quality sequences using BBDuk v38.82 (http://bbtools.jgi.doe.gov). Options used for trimming were: ktrim=r, k = 28, mink = 12, hdist = 1, tbo = t, tpe = t, qtrim = rl, trimq = 20 and minlength = 100. Trimmed reads were assembled using SPADES v3.14[38] with default parameters. Gene prediction using contigs longer than 500 bp was conducted using Prodigal v2.6.3 with default parameters[39]. Genes were annotated using SwissProt and TrEMBL databases[40] with Diamond as the searching tool using default settings[41]. Results were filtered for best match based on bit score, using a minimum threshold of 40% sequence identity and 50% match length. The ANI value between all vs. all genomes was determined using the *ani.rb* script from the Enveomics collection (available at http://enve-omics.gatech.edu)[42]. SNVs detection on assembled contigs was performed using Mauve v2.4.0[43].

Pangenome analysis was applied using gene sequences at the nucleotide level. Predicted gene sequences were compared using an all-versus-all BLASTn v2.2.28[44] to identify the shared reciprocal best matches based on bit score in all pair-wise genome comparisons using a 90% sequence identity cut-off and 90% or more coverage of the query sequence length. The identification of Orthology Groups (OGs), defined as the reciprocal best matches among genomes, was performed using the *ogs.mcl.rb* script from the Enveomics collection[42]. Our core-genome analysis identified 797 shared (core) OGs in single copy (no apparent paralogues). These core-genes were aligned individually using muscle aligner v3.8.31 in order to build consensus trees[45]. For the latter, aligned genes were concatenated using the *Aln.cat.rb* script from the Enveomics collection[42] and phylogenetic analysis was performed using the Neighbor Joining algorithm implemented in ARB v6.0.6[46]. The collection of universal/housekeeping genes were extracted using the *HMM.essential.rb* script from the Enveomics collection[42].

### Metagenome sequencing and analysis

Metagenomes from Mallorca salterns were obtained as detailed in[21]. DNA extraction of the sample collected from the Fuerteventura saltern was performed as detailed in[37]. DNA sequencing libraries were prepared and sequenced as described above for isolate genome libraries. Metagenomics reads were quality-trimmed as described above and subsequently, mapped against all *Sal. ruber* genome sequences competitively (meaning, only the best matching genome among all possible genome matches for each read was retained, and only if above the cut-off for a match; ties for best matches were not counted), using BLASTn v2.2.28[44], to identify reads belonging to each genome or its very close (not sequenced) relatives. After the Blast search, best match read selection (based on bit score) was determined using the *BlastTab.best_hit_sorted.pl* script from the Enveomics script collection[42].

To calculate the total abundance and intra-population diversity of the *Sal. ruber* natural population, we selected reads mapping with sequence identity ≥95% and read coverage ≥70% to any *Sal. ruber* genome in our collection. Subsets of these reads mapping with identity ≥99.3% (using BLASTn[44] as described above; note that 1 mismatch in reads of 150 bps in length results in 99.3% identity) were selected to calculate the abundance of individual isolates or genomovars. The total number of reads mapping to any *Sal. ruber* genome (at identity ≥95%) were normalized by the total sequencing effort (i.e., total number of reads per metagenome) to obtain the final relative abundance for the *Sal. ruber* population (or species). To calculate the difference in abundance between different isolates or genomovars, the analysis only considered the reads that mapped at identity ≥99.3% to one genome and <99.3% to the remaining genomes when representing different genomovars compared to the former genome. The number of reads mapping to each genome was divided by both genome length and total number of metagenomic reads to provide the final relative abundance for the corresponding genome, and by extension, the genomovar the genome represented.

To calculate the coverage of the total *Sal. ruber* population obtained by the sequencing effort applied, we used the Nonpareil tool v3.303[25] on metagenomic reads mapping with identity ≥95% and ≥70% coverage with any of the genomes in our collection. Options used for Nonpareil analysis were: Nonpareil algorithm (-T) = alignment, maximum number of reads as query (-X) = 1,000,000 and identity threshold to group two reads together (-S) = 0.993.

To calculate the rarefaction curves of the genomovar diversity captured by the sequencing effort applied, metagenomic reads were mapped to *Sal. ruber* genomes preserving all matches with identity ≥99.3%. The mapping file was manipulated to remove one target genome at a time (randomly sorted) while recording the number of unique reads mapping at each step, and this process was repeated 100 times to reduce the impact of randomization on the estimates obtained (below). The number of reads were then expressed as the fraction of the maximum number of reads from the *Sal. ruber* species by dividing the observed counts by the total number of reads mapping to any reference genome with identity ≥ 95%. The logarithm of the number of total (dereplicated) genomes used was then expressed as a function of the fraction of *Sal. ruber* reads captured by the genomes, and a linear regression was determined by unweighted least squares and evaluated using Pearson correlation for the region between 20 and 100 genomes. This trendline was extrapolated to 100% coverage of the genomovar diversity (i.e., all reads from the species) to provide an estimate of the number of genomovars represented in the total sequenced fraction. The linear regression, together with the prediction interval, were calculated with the R package stats v4.0.3. The results are reported in the main text, Fig. 3, and Suppl. Table S2.

### Bootstrapped peak finding on the kernel density estimates to estimate variation and confidence intervals of gaps (or valleys) in the ANI value distribution

We performed a bootstrap resampling analysis to produce estimates and confidence intervals for local minimum and maximum ranges in the ANI distribution. We wrote custom Python code for this task that utilized functions from the NumPy, Pandas, SciPy, Matplotlib, and Seaborn packages[47–54]. For each bootstrap iteration, we randomly sampled with replacement the entire ANI value dataset for all *Sal. ruber* genomes, computed the kernel density estimate across the new distribution (scipy.stats.gaussian_kde, bw_method=0.15), and identified local minimums and maximums (scipy.signal.find_peaks, default settings). We repeated this for 10,000 bootstrap iterations. Fig. S3A shows the results of a single bootstrap iteration and Fig. S3B show the results from all 10,000 bootstrap iterations.

### Reporting summary

Further information on research design is available in the Nature Portfolio Reporting Summary linked to this article.

## Data availability

Genomes and metagenomes sequenced for this study from Mallorca and Fuerteventura solar salterns have been deposited in the European Nucleotide Archive (ENA) under BioProject accession numbers PRJEB27680 and PRJEB45291, respectively (Sup. Data S6). Source data are provided with this paper.

## Code availability

All scripts mentioned above are available as part of the Enveomics collection (available at http://enve-omics.gatech.edu)[42], with the exception of the code for ANI peak finding and the kernel density estimate that can be found at: https://github.com/rotheconrad/bacterial_strain_definition. The custom code developed as part of this study to estimate the number of genomovars based on metagenomic datasets is available in GitHub: https://github.com/TomeuViver/Estimations-genomovars.

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

## Acknowledgements

The authors would especially like to thank the whole team at Salinas d'Es Trenc and Gusto Mundial Balearides, S.L. (Flor de Sal d´Es Trenc) and of the Salinas de Fuerteventura (Salinas del Carmen) for allowing access to their facilities and their support in performing the experiments. This study was funded by the Spanish Ministry of Science, Innovation and Universities projects PGC2018-096956-B-C41 and PGC2018-096956-B-C43, RTC-2017-6405-1 and PID2021-126114NB-C42 and PID2021-126114NB-C44, which were also supported by the European Regional Development Fund (FEDER). The research at the IMEDEA was carried out within the framework of the activities of the Spanish Government through the "Maria de Maeztu Centre of Excellence" accreditation to IMEDEA (CSIC-UIB) (CEX2021-001198). KTK's research was supported, in part, by the U.S. National Science Foundation (Award No. 1831582 and No. 2129823). RRM acknowledges the financial support of the sabbatical stay at Georgia Tech supported by the grant PRX18/00048 of the Ministry of Sciences, Innovation and Universities. The computational results presented here have been achieved (in part) using the LEO HPC Infrastructure of the University of Innsbruck. The Fuerteventura samples were taken in accordance with the permit ESNC27, with the unique identifier ABSCH-IRCC-ES-241224-1 that has been provided by the Dirección General de Biodiversidad y Calidad Ambiental del Ministerio para la Transición Ecológica of the Spanish Government. TV acknowledges the "Margarita Salas" postdoctoral grant, funded by the Spanish Ministry of Universities, within the framework of Recovery, Transformation and Resilience Plan, and funded by the European Union (NextGenerationEU), with the participation of the University of Balearic Islands (UIB). TV and RA acknowledge support by the Max Planck Society.

## Author contributions

T.V., Conceptualization, Formal analysis, Investigation, Methodology, Supervision, Validation, Writing – review and editing. R.E.C., Formal analysis, Investigation, Methodology, Software, Validation, Visualization,

Writing – review and editing. L.M.R-R., Investigation, Software, Validation, Writing – review and editing. A.S.R., Resources, Writing – review and editing. J.R-C., Resources, Writing – review and editing. M.L., Resources, Writing – review and editing. S.N.V., Investigation, Validation, Writing – review and editing. R.A., Resources, Supervision, Writing – review and editing. K.T.K., Conceptualization, Resources, Supervision, Writing – review and editing. R.R-M., Conceptualization, Resources, Supervision, Writing – review and editing.

## Funding

## Competing interests
The authors declare no competing interests.

## Additional information

[1]Marine Microbiology Group, Department of Animal and Microbial Biodiversity, Mediterranean Institute for Advanced Studies (IMEDEA, CSIC-UIB), Esporles, Spain. [2]Department of Molecular Ecology, Max Planck Institute for Marine Microbiology, Bremen, Germany. [3]School of Civil and Environmental Engineering, and School of Biological Sciences, Georgia Institute of Technology, Atlanta, GA, USA. [4]Department of Microbiology, and Digital Science Center (DiSC), Universität of Innsbruck, Innsbruck, Austria. [5]Unidad de Epidemiología y Medicina Preventiva, IUSA, Facultad de Veterinaria, Universidad de Las Palmas de Gran Canaria, C/Trasmontaña s/n, Arucas, 35413 Canary Islands, Spain. [6]Department of Biochemistry, Genetics and Microbiology, and Forestry and Agricultural Biotechnology Institute (FABI), University of Pretoria, Pretoria, South Africa. [7]Department of Mathematics and Computer Science, University of the Balearic Islands, Palma 07122, Spain. ✉e-mail: bviver@mpi-bremen.de; kostas.konstantinidis@gatech.edu; ramon@imedea.uib-csic.es

