## [Peer Review File · Nature Communications]

Towards estimating the number of strains that make up a natural bacterial populationReviewer #1 (Remarks to the Author):

In this manuscript Viver et. al. conduct a detailed population genomic analysis of *Salinibacter ruber* isolates. This analysis was conducted for longitudinal data and included consideration of both average nucleotide identity (ANI) and shared genomic content. The overarching questions of the manuscript are determining the number of strains that make up a bacterial population, and proposing ANI thresholds for various sub-species taxonomic identities.

This manuscript contains many interesting sections, and I particularly liked the analysis displayed in Figure 3. However I feel that many of the core conclusions of the paper are over-extrapolated and over-interpreted. A key limitation is that all data generated in this paper is for a single bacterial species, *Salinibacter ruber*, and the population genomics are no-doubt highly influenced by its unique solar saltern habitat. For this reason and a few others enumerated below, I don't think this data can be used to make assumptions about bacteria as a whole, as is done throughout the manuscript. That said, I believe applying these analyses more widely to other bacterial species would yield interesting scientific insights.

Sincerely,
Matthew R Olm

Major comments:

I am concerned that the RAPD protocol resulted in a biased set of sequenced isolate genomes, which makes it much more difficult to infer data about natural population structures. It really just comes down to the fact that the isolate genomes sequenced were not actually randomly chosen; there was preferential sequencing of isolates from different CVs. This non-random sequencing could result in the appearance of population structures (like ANI gaps) that are the result of the strain selection procedure, not natural bacterial population genomics.

The ANI gap shown in Figure 1 does not appear very strong by my eye. I certainly see what the authors are referring to, but based on my look at the histogram, it seems like it 1) it's really driven by a gap in the "Mallorca vs Fuerteventura" comparisons, and 2) it could be caused by random chance. I suggest that the authors use some sort of statistical test to establish whether the gap is statistically significant, and break out the histogram into three separate histograms (based on the current colors of the points).

The definition of strain at 99.99% ANI seemed rather arbitrary. Is there any reason this value was chosen over, say, 99.9% or 99.999%?

I feel the unique lifestyle of *Salinibacter ruber* (as described on lines 392 - 396) really cautions against extrapolation to other bacteria.

On Figure 3, why do you detect 0 genomes when using less than 40% of the reads? That doesn't make sense with my understanding of the procedure that was performed.

Minor comments:

The first sentence of the intro states that a strain is "the smallest distinguishable unit within species". I am not aware of that definition of "strain"; could you provide a citation for that definition?

On line 112 it's stated "The 207 isolates represented 187 distinct CVs. Of them, 118 isolates, representing 107 CVs, including isolates from the same CV, were subjected to whole-genome sequencing." Could you mention why those numbers of CVs and isolates were chosen for sequencing?

On line 158 it's stated "The ANI gap revealed around 99.6-99.8% ANI (Figure 1) was the only ANI range with such a strong bias in terms of pairwise comparisons not falling in this range". What procedure was used to determine this?

On line 206 it's stated: "we defined the (total) *Sal. ruber* population in a sample as any metagenomic sequence from the sample sharing $\geq 95\%$ nucleotide identity with any genome sequence in our collection." Why was a 95% identity threshold used? Wouldn't a more stringent identity threshold be appropriate?

In the section around lines 230 - 241, it's stated that "We dealt with ties in identity in read mapping by not counting such reads.", and "the isolates CZ13 and CZ27, each assigned to a different single-isolate CV, represented the most abundant genomovars in-situ with 2.86% and 2.55% relative abundance, respectively." Because the genomes in your dataset are so similar, you would expect there to be large numbers of "ties", right? By throwing out ties, aren't you biasing against genomes that are similar to one another? For example: consider genomes A, B, and C, where genomes A and B are 99.9% similar to one another and genome C is 97% ANI from A and B. If they're all truly at 25% of the sample, you would expect large numbers of "tie" reads mapping to A and B, substantially reducing their reported relative abundance, while you would expect very few "tie" reads mapping to C. So in the end this bias would make it artificially appear that C has a higher relative abundance than A and B.

I just want to point out that my favorite part of the manuscript was the section from lines 294 - 299; a very neat conclusion and analysis.

Reviewer #2 (Remarks to the Author):

Through the comparison of 138 genomes from *Salinibacter ruber* isolates, the authors assert that the observed gap in the distribution of Average Nucleotide Identity (ANI) represents an inherent genetic boundary within the species. This constitutes the main conclusion drawn in this manuscript. However, the findings presented within the manuscript clearly contradict this central assertion (as outlined in point 2 below). Additionally, it is worth noting that employing metagenomic read mapping can offer more robust insights into whether this gap is natural or not. Intriguingly, although the authors have collected metagenomic sequence data and are familiar with such analyses, the outcomes of this pivotal analysis are conspicuously absent (as indicated in point 3).

My main criticisms are:

1. Lines 49-52. The existence of a universal genetic boundary between named bacterial species and the applicability of ANI thresholds for demarcating bacterial species have been subjects of an ongoing debate in the scientific literature. Notably, one of the co-authors has actively participated in this discussion. It is disappointing that the manuscript presented a very biased review of the current literature, as it failed to recognize the existence of an active debate surrounding this topic.
2. The authors conducted pair-wise comparisons of the genomes of 138 isolates and observed a gap in the distribution ANI values between 99.6% and 99.8%. They argued that this gap is a natural occurrence and not a result of cultivation biases (lines 161-163), which is a key conclusion in this manuscript. However, this conclusion is flawed as it contradicts the subsequent results presented in the section titled "Relative abundance of isolates in the samples of origin reveals cultivation biases" (lines 200-250). In this section, the authors demonstrated that the most frequently cultured genomovars CV1/CV2 (comprising 11 isolates) were actually the least abundant in the natural population, indicating a clear cultivation bias.

The authors reported that they cultured 11 isolates from the CV1/CV2 group and 9 isolates from the CV5/CV6 group. If there is no cultivation bias, one would expect no more than one isolate to be cultured from either group, as neither group is the most abundant in the natural population, and all other CV types have only one cultured isolate. In a scenario when cultivation bias is absent,

only a few pairs of isolates would exhibit ANI values greater than 99.8%. As a result, the tiny peak observed at ANI>99.8% would disappear, effectively eliminating the gap. In other words, the gap appears to be an artifact resulting from cultivation bias.

3. While not entirely devoid of bias, metagenomic sequencing is generally considered less biased than cultivation-based methods due to its culture-independent nature. Consequently, utilizing metagenomic sequencing to sample is considered a better approach to assess the presence of genetic boundaries (gaps) within species. To achieve this, a recommended strategy involves mapping metagenomic reads to a reference genome and subsequently analyzing the distribution of sequence similarities. The authors should be well-acquainted with this methodology, particularly since one of the coauthors previously employed it to explore the concept of microbial species boundaries (Caro-Quintero A and Konstantinidis KT. "Bacterial species may exist, metagenomics reveal." *Environ Microbiol.* 2012 Feb;14(2):347-55). Given that the authors have already amassed metagenomic sequences in the current study, it is perplexing why the results of such an analytical approach were not included in this manuscript.

4. Lines 301-325. While RAPD can serve as a dependable technique for identifying redundant genomes in *S. ruber*, I would caution against the authors drawing broad and generalized conclusions solely based on this one study. It's important to note that the effectiveness of RAPD is contingent on the genetic diversity of the marker genes being employed.

5. Notably, the authors have not supplied the essential scripts or data required for independent verification or validation of the results.

Reviewer #3 (Remarks to the Author):

This paper is an important effort to investigate how much sampling is needed when studying within-species diversity with cultivation and culture independent methods. It also attempts to address an important question whether there is a "natural boundary" in within-species diversity. I would suggest to explore in more depth the properties of the identified ANI gap in within-species pairwise comparisons.

I recommend publication after major revision; below points I would like to bring up for addressing:

Comments/suggestions:

1. It is necessary to statistically test (e.g. Fisher test or something else more suitable) that the number of the within species comparisons at ANI gap (99.2% and 99.8%) is indeed lower than expected by chance in other ANI ranges.

2. There are >200 public isolates of *S. ruber*. Is it possible that they would fill the within-species ANI gap? Or do they bring any extra diversity and more clades, or do they belong to the same subpopulations described in this study?

3. How different is within-species ANI gap when it is only calculated on core genes compared to ANI on any shared regions between each pair of genomes?

4. While Figure 1 provides a good summary of the main result of the paper, it is still not clear how population structure of the *S. ruber* looks like. I would suggest making a tree with an outgroup with tips colored by locations and phenotypic differences between isolates and show the "cut" line on the tree where are different genomovars (or other within-species groups) located on that tree after applying ANI threshold.

5. Since to some extent phenotypes were measured for some of the genomovars (e.g. Figure 5), how are ecotypes/phenotypes distributed in the phylogenetic tree of *S. ruber*? Are any of the accessory genes or variants in core genome associated with phenotypes? Depending on the outcome of such analysis, it might be possible to reveal what drives the ANI gap.

6. How are ANI values distributed across the genome? Which genes drive the difference between genomes?

7. Figure 4 contains "subspecies" but this category has not being used in the figure. What would subspecies mean in this concept figure, any of the within-species catagories?

8. Supplementary figure S5, was ANI normalized based on gene length to define the same allele?

Reviewer #1 (Remarks to the Author):

In this manuscript Viver et. al. conducts a detailed population genomic analysis of *Salinibacter ruber* isolates. This analysis was conducted for longitudinal data and included consideration of both average nucleotide identity (ANI) and shared genomic content. The overarching questions of the manuscript are determining the number of strains that make up a bacterial population, and proposing ANI thresholds for various sub-species taxonomic identities.

This manuscript contains many interesting sections, and I particularly liked the analysis displayed in Figure 3. However, I feel that many of the core conclusions of the paper are over-extrapolated and over-interpreted. A key limitation is that all data generated in this paper is for a single bacterial species, *Salinibacter ruber*, and the population genomics are no-doubt highly influenced by its unique solar saltern habitat. For this reason and a few others enumerated below, I don't think this data can be used to make assumptions about bacteria as a whole, as is done throughout the manuscript. That said, I believe applying these analyses more widely to other bacterial species would yield interesting scientific insights.

Sincerely,
Matthew R Olm

Thank you for the comment and encouragement. Your comments have helped us to substantially improve the manuscript and are truly appreciated. In short, we agree with the reviewer and we change the spin of the paper to say that there is a very interesting ANI gap observed in our species of interest, but the users need to examine their species of interest on a case-by-case basis to assess the presence and exact range of ANI gaps. We also provide further explanations why we believe any possible isolation biases are unlikely to have affected our results and provide strong statistical support for the ANI gap observed.

Major comments:

I am concerned that the RAPD protocol resulted in a biased set of sequenced isolate genomes, which makes it much more difficult to infer data about natural population structures. It really just comes down to the fact that the isolate genomes sequenced were not actually randomly chosen; there was preferential sequencing of isolates from different CVs. This non-random sequencing could result in the appearance of population structures (like ANI gaps) that are the result of the strain selection procedure, not natural bacterial population genomics.

We would like to emphasize that our complete "culture" process is based on random selection, with the exception of the final selection of isolate for genome sequencing where we intentionally opted to not sequence many representatives of the same Clonal Variety (CVs) based on RAPD profiles to avoid sequencing many very similar genomes. Specifically, our RAPD analysis identified a total of 9 CVs with more than one representative genome and we sequence more than 1 representative from only four of them and not all representatives available but a randomly chosen subset. This clearly shows that if we had opted to sequence all the genomes from our collection, the ANI gap we observe in Figure 1 would likely be even more pronounced than what is currently observed. Hence, if there are isolation biases, these are likely against the

(existence of) ANI gap rather than favoring it. We have added a couple sentences to make this point clearer (see lines 126 - 132).

Specifically, our culture approach included the random selection of 96 isolates per sample from those growing on the specific media, resulting in a total of 409 isolates as described previously (Conrad et al., 2022). Subsequently, through MALDI-TOF analysis, we identified those isolates that belonged to the species *Salinibacter ruber*, rendering a total of 207 isolates. To further narrow down our selection, we dereplicated all *Salinibacter ruber* isolates using the RAPD approach, yielding a total of 187 unique RAPD profiles, which we referred to as CVs. Due to financial constraints, we selected 112 isolates with different RAPD profiles for genome sequencing (Conrad et al., 2022). It's important to note that even this selection process remained random. In this manuscript, the only "non-random" step was the selection of a few genomes from 4 CVs for offering intra-CV resolution to some extent (e.g., we did not expect to observe an ANI gap initially to sequence more representative of the same CV!). Clarification in main text can be found in line 124 – 130 and in the discussion (line 376-382).

The ANI gap shown in Figure 1 does not appear very strong by my eye. I certainly see what the authors are referring to, but based on my look at the histogram, it seems like it 1) it's really driven by a gap in the "Mallorca vs Fuerteventura" comparisons, and 2) it could be caused by random chance. I suggest that the authors use some sort of statistical test to establish whether the gap is statistically significant, and break out the histogram into three separate histograms (based on the current colors of the points).

As recommended by the reviewer, we performed a bootstrap resampling analysis to produce estimates and confidence intervals for local minimum and maximum ranges in the ANI distribution using the kernel density estimate. The analysis revealed that within the range of 99.6 to 99.8% ANI values, we observed the lowest number of ANI pairwise comparisons after conducting 10,000 bootstraps, signifying statistical significance. This analysis has been integrated into the main text between lines 157 and 169, and it is also represented in Supplementary Figure S3 (included also below for convenience).

Supplementary Figure S3: Bootstrap resampling analysis to identify variation in local minimum and maximum regions of the ANI distribution using all *Sal. ruber* genomes from this study and from public databases. We performed a bootstrap resampling analysis to produce estimates and confidence intervals for local minimum and maximum ranges in the ANI distribution as described in detail in the Supplemental Material and Methods Section. Briefly, we performed 10000 bootstrap iterations, and for each iteration, we randomly sampled with replacement the entire number ANI pairwise comparisons (i.e. ANI values), computed the kernel density estimate across the new distribution (`scipy.stats.gaussian_kde, bw_method=0.15`), and identified local minimums and maximums (`scipy.signal.find_peaks, default settings`). The top panel shows the empirical distribution for all data (all iterations combined) with the local minimum identified in the range within 99.6% to 99.8% ANI. The second panel shows the results of a single bootstrap iteration and the third panel shows the results from all 10,000 bootstrap iterations with the 95% confidence interval surrounding the mean kernel density estimate in blue. The density and spread of local minimum and maximum values are marked with dark gray or light gray vertical lines in the third panel as well. The bottom panel shows minimum and maximum results from the third panel as a histogram. Note that the empirical distribution in the top panel is far from a Gaussian or Uniform distribution as there is clear deviation from the mean bin count across the ANI distribution. Note the clearly observed minimum around 99.6% to 99.8% ANI is highly stable and consistent with what is reported in the main text.

Following reviewer's recommendation, we have also partitioned the histogram of the key Figure 1 according to the place origin of each genome (included below for convenience). Our observation indicates that there is a discernible ANI gap when comparing genomes originating from the same location, but not when comparing those from different locations. This discrepancy arises because not many genomes from different locations are assignable to same genomovar or strain; thus, it is lack of enough data to evaluate the gap rather than the gap is absent (the gap is even more pronounced among genomes from the same location). Results indicated between lines 143 and 150 of the main text.

The definition of strain at 99.99% ANI seemed rather arbitrary. Is there any reason this value was chosen over, say, 99.9% or 99.999%?

The reviewer is right that the ANI is arbitrary, mostly because there is no other ANI gap within the species to equate to the strain level. But it is not completely arbitrary rather is driven by the level of diversity or noise we see within CVs and among replicate assemblies of the same isolate data, respectively. Specifically, when assessing the ANI values among members of the same CVs (strain), we observed that the lowest ANI value recorded was 99.994 (e.g., ANI value between isolates CZ14 and CZ20). Similarly, independent assemblies of the same isolate data

provided genomes that were >99.99% ANI to each other. Consequently, we established the threshold of 99.99% ANI as the criterion for strain definition and we have explained this point more in our Discussion (specified in lines 396-399 of the main text).

I feel the unique lifestyle of *Salinibacter ruber* (as described on lines 392 - 396) really cautions against extrapolation to other bacteria.

We partly disagree with the reviewer. We believe the patterns observed are not due to the conditions at the salterns or that *Sal. ruber* has unique lifestyle. In other words, these organisms are not “outliers” but simply well-adapted to their respective environment. In this section, we did not aim to report, in detail, on the specific lifestyle characteristics for *Sal. ruber*. Instead, we suggest that the high intra-species diversity may be attributed to the brine inputs utilized for pond replenishment during the summer salt production. And, we highlight that the distinctive trait of *Sal. ruber* lies in its ability to thrive (or simply survive) in different salinity conditions, while its level of intra-species genetic/gene content diversity resembles that of other bacteria found in different ecological niches, characterized by an aerobic heterotrophic lifestyle. We have added a couple sentences to make this point clearer in the discussion section (line 448-450). Furthermore, we have conducted a comprehensive assessment of the ANI gap in a large set of bacterial species, as documented in Rodriguez-R et al. (currently under review and available on BioRxiv, with citations provided in this manuscript), and observed similar patterns.

On Figure 3, why do you detect 0 genomes when using less than 40% of the reads? That doesn't make sense with my understanding of the procedure that was performed.

There appears to be a misunderstanding in the interpretation of the graph. The graph reveals that when one genome is used to recruit metagenomic reads at a 99.3% identity, it successfully recruits an average of 40% of metagenomic reads originating from the natural *Salinibacter ruber* population. The minimum value displayed on the y-axis corresponds to 1 (represented as 100 on the graph).

Minor comments:

The first sentence of the intro states that a strain is “the smallest distinguishable unit within species”. I am not aware of that definition of “strain”; could you provide a citation for that definition?

We have included the citations in the introduction and additional text for clarification (lines 52-26).

On line 112 it's stated “The 207 isolates represented 187 distinct CVs. Of them, 118 isolates, representing 107 CVs, including isolates from the same CV, were subjected to whole-genome sequencing.” Could you mention why those numbers of CVs and isolates were chosen for sequencing?

We have addressed this point above and added more explanations in the revised text (line 126).

On line 158 it's stated "The ANI gap revealed around 99.6-99.8% ANI (Figure 1) was the only ANI range with such a strong bias in terms of pairwise comparisons not falling in this range". What procedure was used to determine this?

We have incorporated a new bootstrapped statistical test and the kernel density estimate to statistically assess the ANI gap between 99.6% and 99.8%. Our response to this comment has been addressed in our reply to the second "major comment." You can find comprehensive details in line 157 of the main text and Supplementary Figure S3.

On line 206 it's stated: "we defined the (total) *Sal. ruber* population in a sample as any metagenomic sequence from the sample sharing $\geq 95\%$ nucleotide identity with any genome sequence in our collection." Why was a 95% identity threshold used? Wouldn't a more stringent identity threshold be appropriate?

In metagenomic studies, we commonly apply the 95% nucleotide identity threshold when recruiting metagenomic reads. This threshold aligns with the established practice in species demarcation, relying on ANI values between genome pairs. Subsequently, we maintain this same identity percentage for quantifying relative species abundance through read recruitment within metagenomes, including the evaluation of the natural *Salinibacter ruber* population.

Furthermore, as can be seen in the revised Figure 1, the genomes used here and represent the dominant *Sal. ruber* population share at last 97.7% ANI. Hence, by using 95% for individual reads we capture well the diversity within the genome (e.g., 97.7% is the mean identity across the genome and several regions show lower identity but in general, not much less than 95% for the orthologous parts). This explanation has been added to the revised text (see lines 233-236).

In the section around lines 230 - 241, it's stated that "We dealt with ties in identity in read mapping by not counting such reads.", and "the isolates CZ13 and CZ27, each assigned to a different single-isolate CV, represented the most abundant genomovars in-situ with 2.86% and 2.55% relative abundance, respectively." Because the genomes in your dataset are so similar, you would expect there to be large numbers of "ties", right? By throwing out ties, aren't you biasing against genomes that are similar to one another? For example: consider genomes A, B, and C, where genomes A and B are 99.9% similar to one another and genome C is 97% ANI from A and B. If they're all truly at 25% of the sample, you would expect large numbers of "tie" reads mapping to A and B, substantially reducing their reported relative abundance, while you would expect very few "tie" reads mapping to C. So, in the end this bias would make it artificially appear that C has a higher relative abundance than A and B.

This is another good point raised by the reviewer but we believe this issue did not affect much the results shown in Figure 3 or 5 because we dereplicated the genomes first at the genomovar level (i.e., ANI values of $\geq 99.6\%$) and the existence of the ANI gap. That is, only one representative genome per genomovar was used and while a couple genomovars may be slightly more closely related to each other, the great majority of genomovars, and thus genomes compared

for these analyses are 97.8-98.5% ANI among them. Accordingly, we have taken no further action to respond to this comment except to specify in the legend of Figure 5 that "each line represents a distinct genomovar".

I just want to point out that my favorite part of the manuscript was the section from lines 294 - 299; a very neat conclusion and analysis.

Thank you for your positive and constructive comments!

Reviewer #2 (Remarks to the Author):

Through the comparison of 138 genomes from *Salinibacter ruber* isolates, the authors assert that the observed gap in the distribution of Average Nucleotide Identity (ANI) represents an inherent genetic boundary within the species. This constitutes the main conclusion drawn in this manuscript. However, the findings presented within the manuscript clearly contradict this central assertion (as outlined in point 2 below). Additionally, it is worth noting that employing metagenomic read mapping can offer more robust insights into whether this gap is natural or not. Intriguingly, although the authors have collected metagenomic sequence data and are familiar with such analyses, the outcomes of this pivotal analysis are conspicuously absent (as indicated in point 3).

My main criticisms are:

1. Lines 49-52. The existence of a universal genetic boundary between named bacterial species and the applicability of ANI thresholds for demarcating bacterial species have been subjects of an ongoing debate in the scientific literature. Notably, one of the co-authors has actively participated in this discussion. It is disappointing that the manuscript presented a very biased review of the current literature, as it failed to recognize the existence of an active debate surrounding this topic.

The debate is mentioned, and a couple relevant publications are cited in the following lines "It is unlikely that this ANI gap is due to cultivation biases because the media and conditions used are thought to be robust for *Sal. ruber* and do not distinguish between members of the species or closely related *Salinibacter* species". In addition, we have added a sentence in the beginning of the Introduction (line 52 to 56) to warn the reader of the debate. "Intermediate identity genotypes, for example, sharing 85–95% ANI, when present, are generally ecologically differentiated and scarcer in abundance, and thus should probably be considered distinct species (Viver 2020, Rodriguez 2021) rather than representing cultivation or other sampling biases (Murray 2021)". However, we did not expand more into this because it is only peripherally relevant to the main thesis of this manuscript.

2. The authors conducted pair-wise comparisons of the genomes of 138 isolates and observed a gap in the distribution ANI values between 99.6% and 99.8%. They argued that this gap is a natural occurrence and not a result of cultivation biases (lines 161-163), which is a key conclusion in this manuscript. However, this conclusion is flawed as it contradicts the subsequent results presented in the section titled "Relative abundance of isolates in the samples of origin reveals cultivation

biases" (lines 200-250). In this section, the authors demonstrated that the most frequently cultured genomovars CV1/CV2 (comprising 11 isolates) were actually the least abundant in the natural population, indicating a clear cultivation bias.

The authors reported that they cultured 11 isolates from the CV1/CV2 group and 9 isolates from the CV5/CV6 group. If there is no cultivation bias, one would expect no more than one isolate to be cultured from either group, as neither group is the most abundant in the natural population, and all other CV types have only one cultured isolate. In a scenario when cultivation bias is absent, only a few pairs of isolates would exhibit ANI values greater than 99.8%. As a result, the tiny peak observed at ANI>99.8% would disappear, effectively eliminating the gap. In other words, the gap appears to be an artifact resulting from cultivation bias.

While there are cultivation biases, as the reviewer also mentions, we do not believe that these have substantially affected our results and thus conclusions for the following reasons. Our isolates represent mostly the abundant members of the *Salinibacter ruber* population because with just a handful of isolates (around 100) we are able to recruit 80% or more of the *Salinibacter ruber* total metagenomic population. So, yes, the frequency of isolates within a CV is not a great representative of the relative abundance of the CV *in situ* but overall, the CV represented by our isolates are the most abundant in-situ and those CV not sampled are mostly the rare biosphere within the species. Further and as mentioned above in a related comment raised by reviewer #1, the majority of our "culture" processes are based on random selection, with the exception of the final selection of isolate for genome sequencing where we intentionally opted to not sequence many representatives of the same Clonal Variety (CVs) based on RAPD profiles to avoid sequencing many very similar genomes. We would like to emphasize that our RAPD analysis identified a total of 9 CVs with more than one representative genome and we sequence more than 1 representative from only four of them and not all representatives available but a randomly chosen subset. This clearly shows that if we had opted to sequence all the genomes from our collection, the ANI gap we observe in Figure 1 would likely be even more pronounced than what is currently observed. Hence, if there are isolation biases, these are likely against the (existence of) ANI gap rather than favoring it. We have added a couple sentences to make this point clearer (line 126 – 132 and between lines 376-382).

3. While not entirely devoid of bias, metagenomic sequencing is generally considered less biased than cultivation-based methods due to its culture-independent nature. Consequently, utilizing metagenomic sequencing to sample is considered a better approach to assess the presence of genetic boundaries (gaps) within species. To achieve this, a recommended strategy involves mapping metagenomic reads to a reference genome and subsequently analyzing the distribution of sequence similarities. The authors should be well-acquainted with this methodology, particularly since one of the coauthors previously employed it to explore the concept of microbial species boundaries (Caro-Quintero A and Konstantinidis KT. "Bacterial species may exist, metagenomics reveal." *Environ Microbiol.* 2012 Feb;14(2):347-55). Given that the authors have already amassed metagenomic sequences in the current study, it is perplexing why the results of such an analytical approach were not included in this manuscript.

We have extensively used metagenomes in this study e.g., figures 2, 3 and 5 are all based on metagenomic read mapping, but metagenomic short-read mapping is not appropriate for

assessing intra-population diversity patterns. The challenge when utilizing Illumina short-read metagenomic sequences for identifying the ANI gap at the "genomovar" level stems from the limited length of these reads, averaging around 150 base pairs in our study. In cases where a single mismatch occurs between a read and the genome, the resulting percentage of identity drops to 99.3%, rendering it impractical to discern the ANI gap within the 99.3% to 100% identity range. We have made this clear in the revised text (Line 296-298 main text).

We consider that future investigations could benefit significantly from the application of long-read PacBio/Nanopore metagenomic sequencing, where read lengths typically range between 3,000 and 10,000 nucleotides. This approach may enable us to more effectively identify the "genomovar" gap. Moreover, an even more promising avenue may emerge if, in the future, we can sequence a large number of single-cell genomes. Please note that in our other paper (Rodriguez-R et al.) we have observed the ANI gap based on long-read data from the human gut and ocean environments.

4. Lines 301-325. While RAPD can serve as a dependable technique for identifying redundant genomes in *S. ruber*, I would caution against the authors drawing broad and generalized conclusions solely based on this one study. It's important to note that the effectiveness of RAPD is contingent on the genetic diversity of the marker genes being employed.

We believe that the RAPD analysis currently stands as an efficient method for distinguishing genomes belonging to the same or different clonal varieties (CVs). As demonstrated in this manuscript, we can confidently assert that isolates exhibiting similar RAPD profiles using three different set of PCR primers, also showed nearly identical genome sequences. Also, please note that the RAPD primers employed in this study are designed as "invented" primers, lacking specific matches to any known sequences or gene. Instead, they randomly amplify genome fragments, allowing for versatile and unbiased genomic differentiation, not specific marker genes. Accordingly, we have taken no further action to respond to this comment except to mention that the RAPD methods employs primers that randomly amplify genome fragments, not specific genes for clarity (included in the text, between line 118 and 121).

5. Notably, the authors have not supplied the essential scripts or data required for independent verification or validation of the results.

We believe that the methods section comprehensively provides all the necessary information to reproduce our work and all of our results. The specific scripts used for genomic comparisons in this manuscript can be accessed through the enveomics webpage, and the scripts for mapping reads to genomes (including tie resolution) are available on the GitHub platform (<https://github.com/rotheconrad>), as also cited in Conrad et al., 2022. Accession codes for the genomes can be found in Supplementary Spreadsheet S5, while the metagenomes used are those published in Conrad et al., 2022, for additional reference and specifics.

Reviewer #3 (Remarks to the Author):

This paper is an important effort to investigate how much sampling is needed when studying within-species diversity with cultivation and culture independent methods. It also attempts to

address an important question whether there is a “natural boundary” in within-species diversity. I would suggest to explore in more depth the properties of the identified ANI gap in within-species pairwise comparisons.

I recommend publication after major revision; below points I would like to bring up for addressing:

Comments/suggestions:

1. It is necessary to statistically test (e.g. Fisher test or something else more suitable) that the number of the within species comparisons at ANI gap (99.2% and 99.8%) is indeed lower than expected by chance in other ANI ranges.

In the manuscript, we performed a bootstrap resampling analysis and employed the kernel density estimator to produce estimates and confidence intervals for local minimum and maximum ranges in the ANI distribution. The analysis conspicuously revealed that within the ANI range of 99.6% to 99.8%, the minimum number of ANI pairwise comparisons was observed. We have comprehensively addressed this issue in the main text, specifically between lines 157 and 169, with the results documented in Supplementary Figure S3 for reference. See also our response to a related comment by reviewer #1.

2. There are >200 public isolates of *S. ruber*. Is it possible that they would fill the within-species ANI gap? Or do they bring any extra diversity and more clades, or do they belong to the same subpopulations described in this study?

Initially, we decided against using all *Salinibacter ruber* genomes available in the public databases due to the absence of identified genomes belonging to the same strain across different samples and the limited number of genomes belonging to the same genomovar. However, in response to the comment by the reviewers, we decided to include a comparison encompassing all available *Salinibacter ruber* genomes in the public databases. The results of statistical tests confirm the persistence of the ANI gap within the 99.6% to 99.8% value range. We have provided this additional information in the main text (between lines 166 and 169) and as Supplementary Figure S5.

Moreover, we have included the statistical bootstrap resampling analysis to produce estimates and confidence intervals for local minimum and maximum ranges in the ANI distribution. The results definitely indicated that the estimated minimum interval of ANI values is located in the range of 99.6 to 99.8% ANI values and the gap did not become less pronounced with the addition of these genomes. We have included this analysis in the main text (line 166 and 169 and Supplementary Figure S5).

Supplementary Figure S5: Bootstrap resampling analysis to identify variation in local minimum and maximum regions of the ANI distribution using all *Sal. ruber* genomes from this study and from public databases (211 genomes). We performed a bootstrap resampling analysis to produce estimates and confidence intervals for local minimum and maximum ranges in the ANI distribution as described in detail in the Supplemental Material and Methods Section. Briefly, we performed 10000 bootstrap iterations, and for each iteration, we randomly sampled with replacement the entire dataset of ANI values, computed the kernel density estimate across the new distribution (`scipy.stats.gaussian_kde, bw_method=0.15`), and identified local minimums and maximums (`scipy.signal.find_peaks, default settings`). The top panel shows the empirical distribution for all data (all iterations combined) with the local minimum identified in the range within 99.6% to 99.8% ANI. The second panel shows the results of a single bootstrap iteration and the third panel shows the results from all 10,000 bootstrap iterations with the 95% confidence interval surrounding the mean kernel density estimate in blue. The density and spread of local minimum and maximum values are marked with dark gray or light gray vertical lines in the third panel as well. The bottom panel shows minimum and maximum results from the third panel as a histogram. Note that the empirical distribution in the top panel is far from a Gaussian or Uniform distribution as there is clear deviation from the mean bin count across the ANI distribution. Note also that the clearly observed minimum around 99.6% to 99.8% ANI is highly stable and consistent with what is reported in the main text and in Figure 1 based only on the genomes reported by our study.

3. How different is within-species ANI gap when it is only calculated on core genes compared to ANI on any shared regions between each pair of genomes?

In response to the reviewers' request, we have computed the ANI values between genome pairs using a set of 793 core genes only. In this instance, we have presented the results in the form of a histogram, as all genomes in the dataset share the complete set of core genes, rendering a two-dimensional plot of ANI values versus percentage of shared genome impractical.

The results reveal that the ANI values between genome pairs are higher when based on the core genes compared to the assessment using complete genomes. Furthermore, both the statistical analysis and the histogram confirm the presence of an ANI gap at the 99.8% value. We have provided the histogram as Supplementary Figure S4 and the text between line 165 and 166. Thus, the ANI gap is shifted a bit upwards when only the core genes are used, as expected because these genes tend to be more conserved, at the sequence level, than the genome average but the gap remains robust.

Supplementary Figure S4: ANI value distribution of *Sal. ruber* genomes based on 793 core ortholog genes. The histogram shows the number of datapoints for x-axis (in 0.1% windows or bins). Contrasting this graph to the main graph shown in Figure 1 shows that ANI values calculated based on the whole genome is similar to that based on core genes only (this Figure).

4. While Figure 1 provides a good summary of the main result of the paper, it is still not clear how population structure of the *S. ruber* looks like. I would suggest making a tree with an outgroup with tips colored by locations and phenotypic differences between isolates and show the “cut” line on the tree where are different genomovars (or other within-species groups) located on that tree after applying ANI threshold.

Supplementary Figure S8 presents a phylogenetic tree based on the 793 core genes alongside a dendrogram reflecting clustering based on the percentage of shared alleles. In this updated version of the figure, we have introduced green fonts for (the ID of) genomes originating from Fuerteventura, while the genomes from Mallorca are represented in black. Furthermore,

within the phylogenetic tree, we have included grey boxes and vertical black lines to indicate genomes belonging to the same genomovar (GV), following the reviewer's suggestion. Although delineating by a vertical line all different genomovars in the phylogenetic tree was unfeasible due to the varied level of intra-genomovar diversity, we successfully incorporated a vertical red line in the allele dendrogram to signify the majority of identified genomovars. Additionally, to differentiate between genomes exhibiting increased abundance in high salt or low salt concentrations (as explained in the next comment, #5), we have introduced red (high salinity) and blue (low salinity) circles next to the ID of the genome.

5. Since to some extent phenotypes were measured for some of the genomovars (e.g Figure 5), how are ecotypes/phenotypes distributed in the phylogenetic tree of *S. ruber*? Are any of the accessory genes or variants in core genome associated with phenotypes? Depending on the outcome of such analysis, it might be possible to reveal what drives the ANI gap.

In the new version of Supplementary Figure S8, we have highlighted those genomes representing genomovars with increased abundance during high-salt conditions using red points, and those that increase their abundance under low-salt conditions using blue points. It can be observed that the genomovars adapted to low-salt conditions are distributed across the phylogenetic tree; that is, this trait is not phylogenetically limited to a major clade or two (see line 499-505).

Further, in our recently published work Conrad et al. (2022), we have identified the isolate-specific or rare genes that become abundant when salinity conditions change and are carried by the corresponding genomovars. These genes include, for example, gene sensing the environment and regulating osmolarity within the cell. Therefore, the ecologically differentiated genomovars carry unique gene content that underlies their ecological preference and we have made this clear in the revised manuscript.

6. How are ANI values distributed across the genome? Which genes drive the difference between genomes?

We do not fully understand this comment since ANI is one value, the mean of the genome, and thus does not vary across the genome. Perhaps the reviewer means sequence identity variation across the genome and the answer would be that this will be evaluated in a subsequent study by our team. Previous work by some of us has shown that the divergent genes between the two available *Sal. ruber* genomes at that time are roughly evenly distributed across the genome (Pena et al, 2010), but a closer look at this is pending now that the number of available genomes has increased substantially. There is already a lot of novel information in this manuscript, we believe.

7. Figure 4 contains “subspecies” but this category has not been used in the figure. What would subspecies mean in this concept figure, any of the within-species categories?

Figure 4 exclusively encompasses taxonomic categories for which we can establish thresholds using ANI or AAI values. In this context, the term "subspecies" should encompass the terms genomovar, strain, and clone. Further, we have removed the term subspecies and instead refer to this as “intra-species units” to avoid confusion because, as the reviewer mentions, subspecies is a category within species itself (and one that most of us do not see a good use for but decided to not discuss this category further in the text).

8. Supplementary figure S5, was ANI normalized based on gene length to define the same allele?

In our allele analysis and the clustering of isolates represented on the heatmap (in the new version indicated as Supplementary Figure S7), we did not consider the normalization of the percentage of identity between gene pairs based on gene length. Our approach solely considered the percentage of identity between the gene shared fraction. However, it's noteworthy that we

conducted a comparison of gene lengths within the Orthology gene groups of the core genes. We observed that the majority of these genes exhibited similar lengths, around 1Kbp. Any potential influence of gene length discrepancies should be minimal, if any, as it would affect only a small subset of genes and is unlikely to significantly impact the clustering of genomes.

Reviewer #1 (Remarks to the Author):

The authors have done an exemplary job modifying the manuscript in response to reviewer comments and I believe it now presents a more balanced view that is ready for publication.

My only remaining suggestion is to change the x-axis label on Figure 3 to "Fraction of metagenomic reads mapped to *Sal ruber* genome(s)" or "Fraction of *Sal ruber* reads mapped" or something like that. The word "used" is ambiguous and I found it confusing.

Sincerely,
Matthew R Olm

Reviewer #2 (Remarks to the Author):

I appreciate the author's effort in responding to my comments. However, as outlined in my responses below, it is clear that the authors have not adequately addressed the central concerns I initially raised in my review of the manuscript.

Comment 1:

In response to my initial critique, the authors assert that "Intermediate identity genotypes, for example, sharing 85–95% ANI, when present, are generally ecologically differentiated and scarcer in abundance, and thus should probably be considered distinct species, rather than representing cultivation or other sampling biases." This statement misinterprets the literature cited in the response, specifically Murray (2021). The Murray et al. study questions using a universal 95% ANI cutoff to define bacterial species and, more importantly, the statistical methodology used to establish such an ANI threshold. It is emphasized that cultivation or sampling bias in bacterial genome sequencing efforts can result in an artificial gap in ANI distribution, which serves as the foundation for defining bacterial species based on ANI. The same concept and statistical approach were applied in this study to define genomevars, very likely leading to a similar erroneous conclusion. Consequently, I strongly disagree with the authors' response that considers this issue as "only peripherally relevant to the main thesis of this manuscript."

Furthermore, I have a concern with the sentence in their response that states, "It is unlikely that this ANI gap is due to cultivation biases because the media and conditions used are thought to be robust for *Sal. ruber* and do not distinguish between members of the species or closely related *Salinibacter* species." Given that the authors have dedicated an entire section of the Results, titled "Relative abundance of isolates in the samples of origin reveals cultivation biases" (lines 225-282), to the discussion of cultivation biases, it is puzzling why they assert that there is no cultivation bias due to the robustness of the media and conditions used for *Sal. Ruber*. This statement also directly contradicts their own findings and conclusions presented in the manuscript at various points. For instance, in their abstract, it is stated that "These data also revealed that the most frequently recovered isolate in lab media was often not the most abundant genomovar in situ, suggesting that cultivation biases are significant, even in cases where cultivation procedures are thought to be robust."

Comment 2:

In response to my second critique, the authors acknowledged the existence of cultivation biases in their study. However, they argued that these biases do not significantly impact their results, as they can recruit 80% or more of the *Salinibacter ruber* metagenomic population with just a limited number of isolates (around 100). Nevertheless, I'd like to point out that recovering 80% of the population does not necessarily imply the absence of substantial biases. For instance, it's conceivable that 10 isolates cover 78% of the population, while the remaining 90 isolates account for just 2%. In such a scenario, we cannot reasonably assert that there are no substantial biases. The authors have not demonstrated that cultivation biases observed in their study will not create

an artificial ANI gap.

The authors responded by stating, "This clearly shows that if we had chosen to sequence all the genomes from our collection, the ANI gap we observe in Figure 1 would likely be even more pronounced than what is currently observed." I concur with the authors that sequencing all the isolates would exacerbate bias and potentially lead to a more pronounced ANI gap. However, the authors went a step further, concluding that, "Hence, if there are isolation biases, these are likely against the (existence of) the ANI gap rather than favoring it." I find the authors' reasoning in this regard to be preposterous and flawed. Using an approach that is less appropriate than the current one does not validate or justify the correctness of the current approach.

Furthermore, the authors mentioned that their "culture" processes are based on random selection, with the exception of the final selection of isolates for genome sequencing. They intentionally chose not to sequence many representatives of the same Clonal Varieties (CVs) based on RAPD profiles to prevent the sequencing of very similar genomes. This rationale, in my opinion, contains a conceptual error. The authors suggested that their "culture" processes, involving random selection of colonies during isolation, reduce or eliminate cultivation biases. However, random colony selection does not necessarily eliminate pre-existing cultivation biases. In essence, randomly sampling a collection that is already biased (colonies) still result in a biased sample.

Therefore, my concerns regarding the presence of cultivation biases in the study and their potential impact on their core conclusion (the existence of an ANI gap) remain valid.

Comment 3:

The authors has addressed my 3rd comment.

Comment 4:

While RAPD may be effective for *Salinibacter ruber*, it's essential to remember that this is a single-species case study, and we cannot draw broad generalizations from the study of one species. I wholeheartedly concur with the observation made by reviewer #1 that many of the paper's core conclusions appear to be overextended and excessively interpreted. Even if the ANI gap is indeed genuine for *Salinibacter ruber*, it remains a representation specific to a single species. Consequently, it does not substantiate the sweeping and all-encompassing conclusions drawn by the authors.

Comment 5:

It will be extremely cumbersome for a reviewer to assemble scripts and data for result reproduction, especially within a two-week review timeframe. Typically, authors provide analysis scripts to facilitate prompt and thorough result replication, which is a standard practice. This collaborative effort is an integral part of the review process, and the recommendations made by the authors seem to needlessly complicate this essential aspect of reviewing.

Reviewer #3 (Remarks to the Author):

Authors have addressed most of my questions/suggestions. I have no further comments.

In the point 6 I meant, for example, applying sliding-window approach(e.g. 1k bp window) to investigate if there are any local higher number of mismatches between strains. But it is indeed a large analysis that might be worth of a separate publication.

Reviewer #1 (Remarks to the Author):

The authors have done an exemplary job modifying the manuscript in response to reviewer comments and I believe it now presents a more balanced view that is ready for publication.
>thank you for your comments and encouragement (on this but also the previous round).

My only remaining suggestion is to change the x-axis label on Figure 3 to “Fraction of metagenomic reads mapped to *Sal ruber* genome(s)” or “Fraction of *Sal ruber* reads mapped” or something like that. The word "used" is ambiguous and I found it confusing.

>done as suggested. New title reads: “Fraction of total metagenomic reads mapped to *Sal. ruber* genomes used in the analysis (%)”

Sincerely,
Matthew R Olm

Reviewer #2 (Remarks to the Author):

I appreciate the author's effort in responding to my comments. However, as outlined in my responses below, it is clear that the authors have not adequately addressed the central concerns I initially raised in my review of the manuscript.

Comment 1:

In response to my initial critique, the authors assert that "Intermediate identity genotypes, for example, sharing 85–95% ANI, when present, are generally ecologically differentiated and scarcer in abundance, and thus should probably be considered distinct species, rather than representing cultivation or other sampling biases." This statement misinterprets the literature cited in the response, specifically Murray (2021). The Murray et al. study questions using a universal 95% ANI cutoff to define bacterial species and, more importantly, the statistical methodology used to establish such an ANI threshold. It is emphasized that cultivation or sampling bias in bacterial genome sequencing efforts can result in an artificial gap in ANI distribution, which serves as the foundation for defining bacterial species based on ANI. The same concept and statistical approach were applied in this study to define genomevars, very likely leading to a similar erroneous conclusion. Consequently, I strongly disagree with the authors' response that considers this issue as "only peripherally relevant to the main thesis of this manuscript."

>We agree with the reviewer that we had to give more weight and clearer about this alternative explanation. Accordingly, we modified the corresponding sentence to read like this “Intermediate identity genotypes, for example, sharing 85–95% ANI, when present, are scarcer in abundance due to ecological differentiation, and thus should probably be considered distinct species^{4,7} (for a contrasting opinion that attributes such ANI gaps to cultivation or other sampling biases see ⁸)”. We do believe that this alternative hypothesis is much less likely, however, and thus, we

continue to favor our hypothesis for the reasons explained in this manuscript but also in Rodriguez-r et al., Nat. Comms 2021. To briefly summarize the main points here for this reviewer: In their technical comment, Murray and colleagues took the collection of genomes and introduced a biased selection to maximize phylogenetic diversity by subsampling the genomospecies that are most overrepresented. That doesn't represent the natural distributions more faithfully, it simply acts as systematically reduce the rightmost ANI peak. Further, it is highly likely that the low abundance intermediate genotypes, which fall within the ANI gaps, are low abundance due to their ecological differentiation compared to the dominant/abundant genotypes at the time of sampling, and not isolation or sampling biases (so, it may be ecology, not sampling bias that is responsible/driving the gaps). “

Furthermore, I have a concern with the sentence in their response that states, "It is unlikely that this ANI gap is due to cultivation biases because the media and conditions used are thought to be robust for *Sal. ruber* and do not distinguish between members of the species or closely related *Salinibacter* species." Given that the authors have dedicated an entire section of the Results, titled "Relative abundance of isolates in the samples of origin reveals cultivation biases" (lines 225-282), to the discussion of cultivation biases, it is puzzling why they assert that there is no cultivation bias due to the robustness of the media and conditions used for *Sal. Ruber*. This statement also directly contradicts their own findings and conclusions presented in the manuscript at various points. For instance, in their abstract, it is stated that "These data also revealed that the most frequently recovered isolate in lab media was often not the most abundant genomovar in situ, suggesting that cultivation biases are significant, even in cases where cultivation procedures are thought to be robust."

>There are some isolation biases clearly and we believe we are clear about this as also the reviewer implies above. But these seems to be limited to the relative abundance of the abundant genomovars, and do not select for rare genomovars that could possibly have distorted the picture. Hence, we believe that the emerging picture is indeed representative of the major patterns observed within the *Sal. ruber* population and quantitative. We have edited our text to be clearer about this key issue. Please see top paragraph on page 17 (especially lines 5-7).

Comment 2:

In response to my second critique, the authors acknowledged the existence of cultivation biases in their study. However, they argued that these biases do not significantly impact their results, as they can recruit 80% or more of the *Salinibacter ruber* metagenomic population with just a limited number of isolates (around 100). Nevertheless, I'd like to point out that recovering 80% of the population does not necessarily imply the absence of substantial biases. For instance, it's conceivable that 10 isolates cover 78% of the population, while the remaining 90 isolates account for just 2%. In such a scenario, we cannot reasonably assert that there are no substantial biases. The authors have not demonstrated that cultivation biases observed in their study will not create an artificial ANI gap.

The authors responded by stating, "This clearly shows that if we had chosen to sequence all the genomes from our collection, the ANI gap we observe in Figure 1 would likely be even more

pronounced than what is currently observed." I concur with the authors that sequencing all the isolates would exacerbate bias and potentially lead to a more pronounced ANI gap. However, the authors went a step further, concluding that, "Hence, if there are isolation biases, these are likely against the (existence of) the ANI gap rather than favoring it." I find the authors' reasoning in this regard to be preposterous and flawed. Using an approach that is less appropriate than the current one does not validate or justify the correctness of the current approach.

>We believe that we addressed these points above and thus have taken no further action to respond here.

Furthermore, the authors mentioned that their "culture" processes are based on random selection, with the exception of the final selection of isolates for genome sequencing. They intentionally chose not to sequence many representatives of the same Clonal Varieties (CVs) based on RAPD profiles to prevent the sequencing of very similar genomes. This rationale, in my opinion, contains a conceptual error. The authors suggested that their "culture" processes, involving random selection of colonies during isolation, reduce or eliminate cultivation biases. However, random colony selection does not necessarily eliminate pre-existing cultivation biases. In essence, randomly sampling a collection that is already biased (colonies) still result in a biased sample.

Therefore, my concerns regarding the presence of cultivation biases in the study and their potential impact on their core conclusion (the existence of an ANI gap) remain valid.

>We respectfully disagree with the reviewer on their main conclusion here because we are not aware of *Sal. ruber* subpopulations that do not grow in our media.

Comment 3:

The authors has addressed my 3rd comment.

Comment 4:

While RAPD may be effective for *Salinibacter ruber*, it's essential to remember that this is a single-species case study, and we cannot draw broad generalizations from the study of one species. I wholeheartedly concur with the observation made by reviewer #1 that many of the paper's core conclusions appear to be overextended and excessively interpreted. Even if the ANI gap is indeed genuine for *Salinibacter ruber*, it remains a representation specific to a single species. Consequently, it does not substantiate the sweeping and all-encompassing conclusions drawn by the authors.

>We have edited some of the sweeping statements. In addition, we have a paper that is coming out in mBio (cited here) that shows similar patterns in many additional species. Accordingly, we have taken no further action to respond here.

Comment 5:

It will be extremely cumbersome for a reviewer to assemble scripts and data for result reproduction, especially within a two-week review timeframe. Typically, authors provide analysis scripts to facilitate prompt and thorough result replication, which is a standard practice. This collaborative effort is an integral part of the review process, and the recommendations

made by the authors seem to needlessly complicate this essential aspect of reviewing.

>We thank the reviewer for this note, and we have added in the Code Availability section the details on where all of our scripts can be found on GitHub (and are better organized and documented now). Anybody with some basic bioinformatics skill should be able to reproduce our results. Please see Code Availability section for the details.

Reviewer #3 (Remarks to the Author):

Authors have addressed most of my questions/suggestions. I have no further comments.

In the point 6 I meant, for example, applying sliding-window approach(e.g. 1k bp window) to investigate if there are any local higher number of mismatches between strains. But it is indeed a large analysis that might be worth of a separate publication.

>Thank you for this suggestion. It will be included in an upcoming publication by our team shortly. Please stay tuned!